mechanics/materials science/geometry

Riemannian metric, shape, embedded surfaces

**Authors for correspondence:**
Shankar Ghosh
e-mail: sghosh@tifr.res.in
Nitin Nitsure
e-mail: nitsure@math.tifr.res.in

# Moulding three-dimensional curved structures by selective heating

Harsh Jain[1], Shankar Ghosh[1] and Nitin Nitsure[2]

[1]Department of Condensed Matter Physics and Materials Science, and [2]School of Mathematics, Tata Institute of Fundamental Research, Mumbai 400005, India

HJ, 0000-0003-2268-3837; SG, 0000-0003-3617-5390; NN, 0000-0002-6652-7919

It is of interest to fabricate curved surfaces in three dimensions from homogeneous material in the form of flat sheets. The aim is not just to obtain a surface which has a desired intrinsic Riemannian metric, but to get the desired embedding in $\mathbb{R}^3$ up to translations and rotations. In this paper, we demonstrate three generic methods of moulding a flat sheet of thermo-responsive plastic by selective contraction induced by targeted heating. These methods do not involve any cutting and gluing, which is a property they share with origami. The first method is inspired by tailoring, which is the usual method for making garments out of plain pieces of cloth. Unlike usual tailoring, this method produces the desired embedding in $\mathbb{R}^3$. The second method just aims to bring about the desired new Riemannian metric via an appropriate pattern of local contractions, without directly controlling the embedding. The third method is based on triangulation, and seeks to induce the desired local distances. This results in getting the desired embedding in $\mathbb{R}^3$. The second and the third methods, and also the first method for the special case of surfaces of revolution, are algorithmic in nature. We explain these methods and show examples.

## 1. Introduction

Common materials such as steel, paper, plastic and cloth are usually produced as flat sheets. More complicated curved and folded shapes have to be fashioned out of such flat sheets. For example, dresses are tailored for the human form out of a cloth which is flat, or globes of the earth with maps are fashioned from printed flat sheets. Footballs are often made by stitching together a very large number of small flat pentagonal and hexagonal pieces of leather.

All these curved surfaces are made by cutting out various shapes from a flat sheet and then gluing, welding or stitching together some of the resulting pairs of edges. In contrast to this,

in nature there are situations where a surface in $\mathbb{R}^3$ is either generated or gets modified because of local contractions and expansions of a flat sheet or some other prior shape, without any cutting or gluing [1–3]. This raises the question of how to mould a desired curved shape from a flat sheet by using selective local expansion and contraction, but without any cutting and gluing [4–6]. One may compare this question with those approaches of moulding that are inspired by the art of origami, which is to approximate three-dimensional shapes from a flat sheet of paper by folding but without cutting or gluing [7–13] For us, folding is to be replaced by selective local expansions/contractions. Such expansions/contractions appear to be more intrinsic to the surface—and therefore more natural—than folds, as folds need to be implemented from the outside by an external agent.

In this paper, we discuss three methods of making such curved surfaces from a plastic material which contracts on heating and remains contracted after returning to room temperature. It will be clear from §2 that there is no loss of generality in confining ourselves to contractions alone (instead of using both contractions and expansions) because of a certain idea that we call the *c*-trick, which essentially consists of starting with appropriately larger sheets, so that further expansions are not needed, and contractions alone suffice. Similarly, we could have worked with materials which only expand, by a modified *c*-trick which amounts to starting with appropriately smaller sheets so that local expansions alone suffice to get the desired shape. It is also possible to work with materials whose expansions or contractions are temporary, so that the moulded surfaces return to their original flat state after some time. Examples of such materials include liquid crystalline elastomer [4,14], thermo-responsive polymer gels [5,6] and hygroscopic surfaces [15]. In this paper, we use a material that contracts, so we will focus on this case, and not make any more comments about expansions. The first method, which we call the contraction-tailoring method, is directly inspired by the usual tailoring of clothes. The second method, which we call Riemannian metric moulding, endeavours to produce a surface which has a prescribed Riemannian metric. It should, however, be noted that the Riemannian metric on a surface in general *does not* correspond to a unique equivalence class of embeddings of the surface in $\mathbb{R}^3$ up to Euclidean isometries of $\mathbb{R}^3$. This is related to the somewhat subtle issue of rigidity of Riemannian embeddings, which is discussed later. The third method, which we call the shape moulding method, endeavours to produce a surface which has a prescribed shape in $\mathbb{R}^3$, where by shape we mean an equivalence class of embeddings under Euclidean isometric transformations of the ambient $\mathbb{R}^3$. Of course, achieving a desired shape ensures in particular that the desired intrinsic Riemannian metric is obtained. All three methods depend only on contractions, and do not involve any cutting and gluing.

In what follows, we first recall some geometric concepts relevant to the problem. Then we discuss some basic theoretical aspects and limitations of the above three moulding methods. Finally, we report on our practical implementations of these methods where the material is a flat sheet of thermo-responsive plastic which contracts on heating.

Some earlier experiments reported in the literature aimed at obtaining curved surfaces in $\mathbb{R}^3$ from flat surfaces relied on modifying the flat Riemannian metric of the starting planar surfaces [4,5,16–18]. This involved stretching, contracting and rotating pre-designated patches on the starting surface to get the desired new Riemannian metric. However, as we discuss later, this does not uniquely determine the embedding class (shape) of the resulting surface into $\mathbb{R}^3$. While one of our three methods aims to get the target Riemannian metric and has a similar weakness, our other two methods give us better control over the embedding into $\mathbb{R}^3$.

## 2. Geometric aspects of the moulding problem

Let $\mathbb{R}^3$ denote the three-dimensional Euclidean space, with Cartesian coordinates $x$, $y$, $z$. The Euclidean distance between two points $P_1 = (x_1, y_1, z_1)$ and $P_2 = (x_2, y_2, z_2)$ in $\mathbb{R}^3$ is given by the Pythagorean formula $\|P_1 - P_2\| = \sqrt{(x_1 - x_2)^2 + (y_1 - y_2)^2 + (z_1 - z_2)^2}$. A related structure on $\mathbb{R}^3$ is its Riemannian metric, given by the formula $\mathrm{d}s^2 = \mathrm{d}x^2 + \mathrm{d}y^2 + \mathrm{d}z^2$, which measures the squared lengths of infinitesimal displacements of tangent vectors.

If $M$ is a surface embedded in the three-dimensional Euclidean space $\mathbb{R}^3$, then the two kinds of metrics on the ambient $\mathbb{R}^3$ ('distance metric' and 'Riemannian metric') induce corresponding structures on $M$. The Riemannian metric induced on $M$ can be locally expressed as $\mathrm{d}s^2 = E\,\mathrm{d}u^2 + 2F\,\mathrm{d}u\,\mathrm{d}v + G\,\mathrm{d}v^2$ in terms of a local coordinate patch $(u, v)$ on $M$, where $E$, $F$, $G$ are functions of $u$, $v$. For $P, Q \in M$, the induced distance metric $\|P - Q\|$ is simply the straight line distance between $P$ and $Q$ in the ambient $\mathbb{R}^3$ (which may be much shorter than the geodesic distance between these points on $M$).

Our aim is to fashion a surface $M \subset \mathbb{R}^3$ by deforming a flat piece $D$ of plastic, which has its starting intrinsic distance and Riemannian metric induced by its inclusion in the Euclidean plane $\mathbb{R}^2$. Note that $D$ can be any suitable domain in $\mathbb{R}^2$, for example, a disc or a rectangle or an annulus. Such a fashioning corresponds to a sufficiently smooth continuous map $\varphi$ from $D$ into $\mathbb{R}^3$ which maps $D$ homeomorphically onto $M$. Note that such a $\varphi$ is far from unique; that is, if one such $\varphi$ exists, then there are uncountably many other such $\varphi$'s possible. We want a method of moulding $D$ which will, for a desired $M \subset \mathbb{R}^3$ which is abstractly homeomorphic to $D$, first choose a suitable embedding $\varphi : D \to \mathbb{R}^3$ whose image is $M$ (up to an isometry of $\mathbb{R}^3$), and then bring it about physically. Note that the distance metrics of $D$ and $M$ (as subspaces of $\mathbb{R}^2$ and $\mathbb{R}^3$, respectively) are different, and moreover $\varphi$ will not usually carry the intrinsic Riemannian metric of $D$ into that of $M$, though there are exceptional cases such as rolling a flat sheet into a portion of a cone or a cylinder where the distance metric changes but the Riemannian metric remains the same. As our method of moulding is by thermal contraction, it is necessary for us that $\varphi$ should everywhere be a contraction in terms of the original flat Riemannian metric on $D$.

We now precisely formulate the condition that $\varphi : D \to M$ is everywhere a local contraction. Let $X, Y$ be Cartesian coordinates on $D$ and let $x, y, z$ be Cartesian coordinates on $\mathbb{R}^3$. Let

$$\varphi(X, Y) = (\varphi_1(X, Y), \varphi_2(X, Y), \varphi_3(X, Y)) \in \mathbb{R}^3. \tag{2.1}$$

Then the Riemannian metric $dx^2 + dy^2 + dz^2$ on $\mathbb{R}^3$ pulls back under $\varphi$ to the Riemannian metric $E\,dX^2 + 2F\,dX\,dY + G\,dY^2$ on $D$ where

$$E(X, Y) = \left(\frac{\partial \varphi_1}{\partial X}\right)^2 + \left(\frac{\partial \varphi_2}{\partial X}\right)^2 + \left(\frac{\partial \varphi_3}{\partial X}\right)^2,$$
$$F(X, Y) = \frac{\partial \varphi_1}{\partial X}\frac{\partial \varphi_1}{\partial Y} + \frac{\partial \varphi_2}{\partial X}\frac{\partial \varphi_2}{\partial Y} + \frac{\partial \varphi_3}{\partial X}\frac{\partial \varphi_3}{\partial Y}$$
$$\text{and} \qquad G(X, Y) = \left(\frac{\partial \varphi_1}{\partial Y}\right)^2 + \left(\frac{\partial \varphi_2}{\partial Y}\right)^2 + \left(\frac{\partial \varphi_3}{\partial Y}\right)^2. \tag{2.2}$$

The condition that $\varphi$ is a contraction at $(X, Y)$ means both the eigenvalues of $\begin{pmatrix} E & F \\ F & G \end{pmatrix}$ are $\leq 1$ at $(X, Y)$, that is, $(E + G + \sqrt{(E-G)^2 + 4F^2})/2 \leq 1$ at $(X, Y)$.

If an initially chosen mathematical candidate map $\varphi : D \to M$ is not everywhere a contraction, then we can systematically modify $D$ and $\varphi$ by the following trick, which we call the *c-trick*. We first choose a constant $c \geq 1$, such that at any point of $D$, the infinitesimal linear amplification made by the map $\varphi$ is bounded above by $c$, that is

$$\max_D \frac{E + G + \sqrt{(E-G)^2 + 4F^2}}{2} \leq c^2. \tag{2.3}$$

Now let $D^*$ be a new flat piece which is $c$ times $D$ (the linear dimensions of $D^*$ are $c$ times the corresponding linear dimensions $D$). Let the map $\psi : D^* \to D$ be the homeomorphism which is an isotropic contraction by the factor $c$. Let $X^*, Y^*$ be Cartesian coordinates on $D^*$ with $X = X^*/c$, $Y^* = Y/c$. Then $\varphi^* = \varphi \circ \psi$ is everywhere a contraction on $D^*$, as the old $E, F, G$ are now replaced by

$$E^*(X^*, Y^*) = \frac{E(X^*/c, Y^*/c)}{c^2},$$
$$F^*(X^*, Y^*) = \frac{F(X^*/c, Y^*/c)}{c^2}$$
$$\text{and} \qquad G^*(X^*, Y^*) = \frac{G(X^*/c, Y^*/c)}{c^2}. \tag{2.4}$$

With this, we get $\max_{D^*} ((E^* + G^* + \sqrt{(E^* - G^*)^2 + 4F^{*2}})/2) \leq 1$. Thus, replacing the original candidate $(D, \varphi)$ as the starting point for moulding by the pair $(D^*, \varphi^*)$ ensures that the modification is everywhere a contraction.

The possibility of replacing $(D, \phi)$ by $(D^*, \phi^*)$ shows that there is no loss of generality in limiting our methods to contraction alone, without the need for any expansion.

## 2.1. Surfaces of the form $z = f(x, y)$

If a surface $M$ is given by an equation $z = f(x, y)$ defined on $D \subset \mathbb{R}^2$, then the inverse of the vertical projection on the $x,y$-plane gives a function $\varphi : D \to M$. In terms of the induced curvilinear coordinates $x$, $y$ on $M$ the metric on $M$ takes the form $\mathrm{d}s^2 = (1 + f_x^2)\,\mathrm{d}x^2 + 2f_x f_y\,\mathrm{d}x\,\mathrm{d}y + (1 + f_y^2)\,\mathrm{d}y^2$. If $\mathrm{grad}(f)$ is 0 at $(x, y) \in D$, then both the eigenvalues are 1 as the vertical projection is an isometry infinitesimally near the point. In general, the two eigenvalues are 1 and $1 + (f_x)^2 + (f_y)^2 \geq 1$, corresponding to eigenvectors $\mathrm{grad}(f)^{\perp}$ and $\mathrm{grad}(f)$. Hence, in this case, we can take any $c$ such that

$$\max_D (1 + f_x^2 + f_y^2) \leq c^2. \qquad (2.5)$$

The transformations $\varphi : D \to M$ and $\varphi^* : D^* \to M$, in the case where $\varphi$ is the inverse of a vertical projection $M \to D$, are illustrated in figure 1. We have $\varphi^* = \varphi \circ \psi$ where $\psi : D^* \to D$ is the contraction by $c$.

# 3. Moulding by contraction

We use a thermo-responsive polymer sheet commercially known as Shrinky Dink [17,19–21], a material that contracts when heat is applied, as our plain sheet $D$ from which the curved shape $M$ is to be moulded. If heated uniformly by painting it black and exposing it to infrared light for a few minutes, a free-standing piece of this material contracts isotropically by a multiplicative factor $\gamma$ of 0.4, and becomes approximately $6 \sim 1/(0.4)^2$ times thicker. If only a part of a piece of the material is painted black, then the result is more complicated as it depends on the unheated boundary which retains its original length. On heating, a painted strip bends more towards the blackened side which is hotter, just as a bi-metallic strip bends because of differential contraction [19]. The three methods of moulding described here are not particularly limited to the kind of thermo-responsive polymer sheet chosen for the experiments presented in the paper. These methods are general and should also apply to other suitable materials [5,6,13–15,22–27].

In our experiment, the heating responded nonlinearly to the degree of shading intensity, with a negligible response below a certain threshold and a nearly full response above it. This made it more convenient to use a tiled pattern of black and white regions instead of smoothly varying shading. For such tiled patterns to be effective, we found that the black (white) regions should not be too small, otherwise they lose (gain) too much heat to (from) the surroundings.

## 3.1. Experimental details

The thermo-responsive polymer sheets that we used in our experiments were commercially sourced and were of the brand 'Shrinky Dink'. These sheets are 0.25 mm thick and they contract when heated to temperatures greater than 100°C. The three protocols of moulding as described in the paper require us to selectively heat specific portions of the sheet. This was achieved by printing black patches on the sheet using an office laser printer [20]. We used a 150 W infrared incandescent bulb as a heating source. The black patches selectively get hot and contract as they absorb more radiation. To ensure uniform coverage of radiation, the plastic pieces of the material were kept at a distance of ≈16 cm from the bulb, and the pieces were continuously rotated. The distance between the bulb and the piece of the material was suitably chosen to obtain a homogeneous level of radiation which would heat a blackened disc of 10 cm diameter to about 100°C in a few minutes. To avoid the substrate from getting hot, the shrinkable polymer piece was placed on a flat Teflon sheet. Teflon does not absorb the radiation efficiently and hence remains relatively cold (≈45°C). This ensures that there is no significant heating by conduction, which would affect the white (non-printed) parts also. The duration of heating was set by visual inspection of the emerging moulded shape. The temperature of the sheet was monitored using an infrared camera (FLIR A600).

The implementation of the metric moulding and the distance moulding via triangulation involves extended exposure of the material to the radiation. To prevent the white portions from getting soft by thermal conduction we reinforce the white portion by selectively printing a 0.5 mm coating of ABS plastic on it using a three-dimensional printer.

### 3.1.1. Features of the practical implementation

(i) The black regions soften (elastic modulus is of the order of 1 MPa) and contract on exposure to thermal radiation. For our fixed regime of thermal exposure that is detailed above, we define a

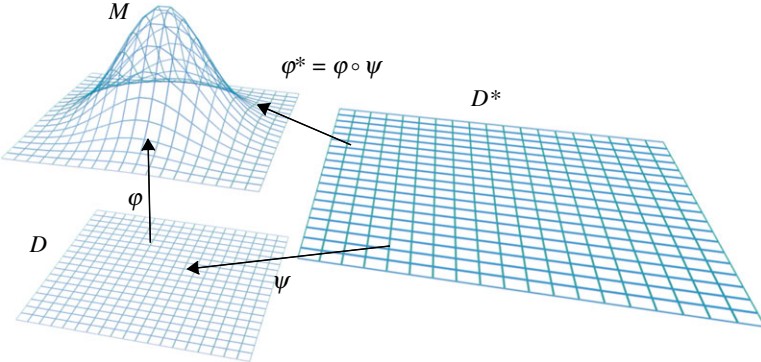

**Figure 1.** The figure shows a schematic of the maps $\varphi : D \to M$, $\psi : D^* \to D$ and the composite map $\varphi^* = \varphi \circ \psi$.

dimensionless quantity $\gamma$, which we call the *contraction coefficient* as the ratio

$$\gamma = \frac{\text{the length of a black region after contraction}}{\text{its original length}} = \frac{b'}{b}. \tag{3.1}$$

The factor $\gamma$ depends on the original length $b$ of the black region. This dependence is graphically depicted in figure 2a. As can be seen from the graph, $\gamma$ is approximately constant $\approx 0.5$ for $b \geq 4$ mm. Below 4 mm, the contraction coefficient approaches 1 because of the heat loss to the neighbouring white region. The exact nature of the curve in figure 2a is dependent on the extent of the white region that surrounds the black region.

(ii) When our experimental protocol was applied to identical pieces of plastic, each of which was uniformly painted to a different degree of blackness varying from white to shades of grey to black, it was observed that the contraction coefficient $\gamma$ responded nonlinearly to the degree of shading intensity, with a negligible response below a certain threshold. This made it more convenient to use a tiled pattern of black and white regions instead of smoothly varying shading. As explained earlier, the black and white regions should not be too small so that the undesired effect of thermal conduction is kept limited.

(iii) On exposure to radiation, the printed side heats more and therefore, whenever possible, the sheet bends towards the printed side much as a bi-metallic strip bends, because of differential contraction. This effect, though unintended, can be put to use as explained in §5. One of the uses is to choose a particular chirality for the moulded shape. Geometrically speaking, a flat plastic disc or rectangle $D$ in $\mathbb{R}^2$ has no physically preferred orientation (chiral structure). On the other hand, a surface $M$ in $\mathbb{R}^3$, though diffeomorphic to $D$, can have a chirality (for example, a rectangular strip can become a winding spiral ramp, which could be right-handed or left-handed). The question arises whether the black and white pattern can be so given to produce the desired chirality. This is indeed possible by selectively painting on one side or other on different locations on $D$ which converts $D$ into an a-chiral object (its mirror image is not obtainable from itself by just a translation and a rotation in $\mathbb{R}^3$—see the appendix in the arXiv version-1 of Ghosh *et al.* [28] for a relevant discussion on chirality. This appendix is not included in the published version of the paper [29]).

(iv) Because of imperfections and lack of uniformity of heating, it can happen that chiral symmetry can get broken in unintended ways, which we may call a spontaneous breaking of chiral symmetry. An example of this is shown in figure 2c. The figure shows a twisted shape that is generated by heating a strip with a pattern that is shown in the inset of the figure. The sense (chirality) of the twists are not determined by the pattern of blackening, but arises out of spontaneous breaking of chiral symmetry.

(v) It is not desirable to have a large black region surrounded by a white region, as the middle of the black region thins on heating, with the material migrating to the boundary. This happens because the temperature in the central part of the black region becomes higher making the material there softer, and therefore susceptible to the contracting elastic pull exerted by the boundary which is anchored to the surrounding colder and hence more rigid white region. The thermal images and the temperature profiles that bear the above point are shown in figure 2e,f. An extreme example of this phenomenon is that when subjected to overheating induced by prolonged exposure, a mechanical tear develops in the middle of a black region

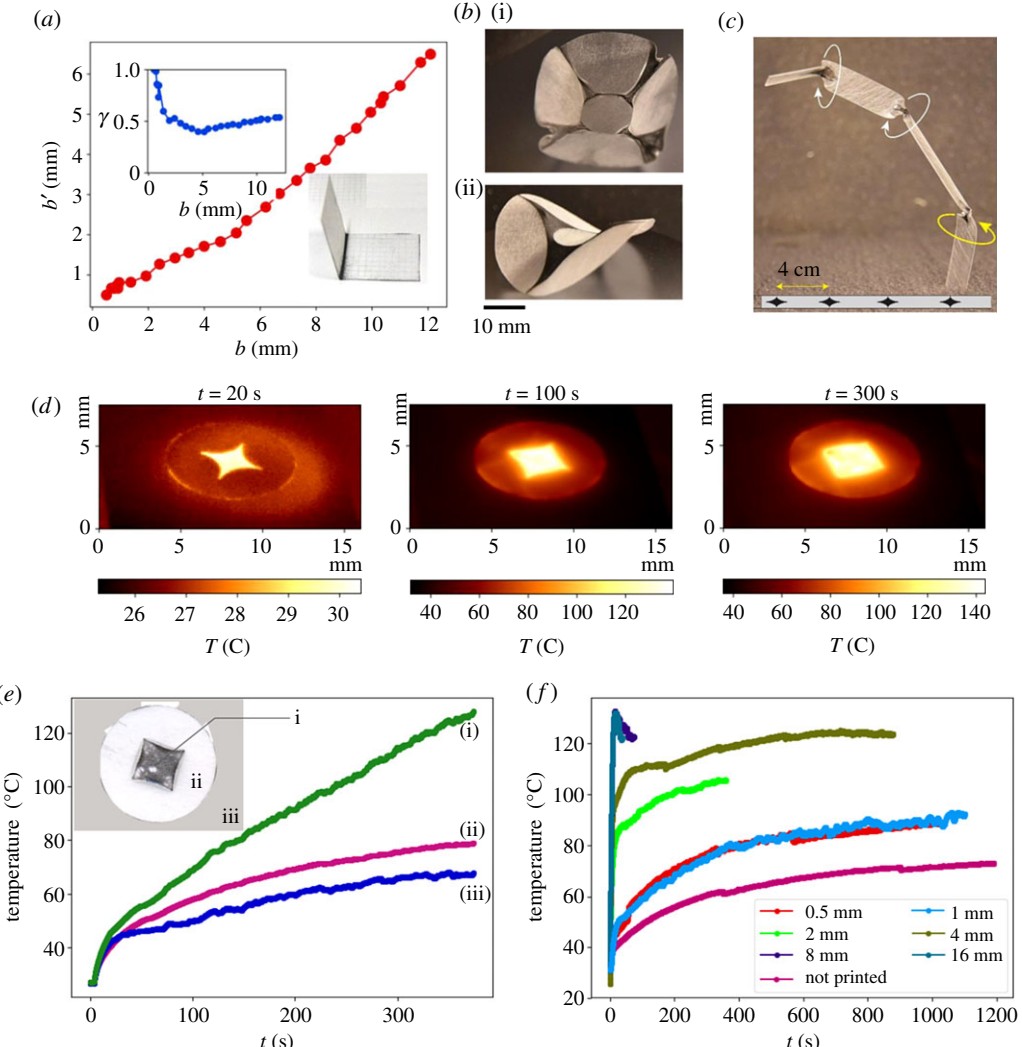

**Figure 2.** Panel (*a*) shows the variation of the width $b'$ of a black strip after contraction by heating, as a function of its initial width $b$. The inset in the top shows the variation of the contraction coefficient $\gamma = b'/b$ as a function of $b$. The bottom inset to (*a*) shows that on heating the plastic bends towards the black region. The panel (*b*) show that the deformations do not penetrate much into the closed white regions that are entirely surrounded by black regions. (*c*) The figure shows the twisting of a 1 cm wide strip of plastic on which the pattern in the inset is printed (entirely on one side). While the printed strip is achiral, on heating it gets twisted into a structure in which the sense of the twist (marked by the circular arrows) changes along the length, demonstrating a spontaneous breaking of chiral symmetry. The panels in (*d*) show the thermo-graphs of a printed disc (6 cm in diameter) of plastic for different durations of heating. These thermographs were obtained using a infrared camera (FLIR A600). (*e*) The graphs show the variation of temperature as a function of time for locations on the disc marked by (i), (ii) and (iii) in the inset. The green line and the magenta line, respectively, show the temperature variation at the centre of the black printed region and in the white region. The blue line shows the temperature variation of the Teflon piece on which the material is kept. The photograph in (*e*) shows the deformation and rupture of a black region that is completely surrounded by a white region, when exposed to infrared light. (*f*) The figure shows the temperature variation in the centre of the black patches of different sizes.

which is surrounded by a white region. This can be seen in the inset of figure 2*e*, in which the black material has moved closer to the nearest edge, leading to the creation of multiple thick and thin regions. Prior to overheating, the sheet develops a small negative Gaussian curvature, which disappears when the centre of the black region develops some tears.

(vi) While being heated under our experimental protocol, the temperature at a point on the sheet decreases as we move away from the black region into the white region. It drops below 90°C in about 4 mm from the boundary of the black region. There is no discernible contraction at temperatures below 90°C, so as one moves away from the black region into the white region, the contraction coefficient rises from 0.5 to 1 within 4 mm. This tells us that to be a non-

contracting region, the width of a white patch or strip which has a large neighbouring black region has to be considerably more than 4 mm. However, this limitation can be overcome by coating the white region by a rigid material before heating. In our experiments, we have used a 0.5 mm coating of ABS plastic as the rigid material.

(vii) If the boundary of a small closed region is darkened but its interior is kept white, then even after heating the interior region remains flat w.r.t. the ambient Euclidean 3-space, while the exterior may acquire a curvature w.r.t. the ambient Euclidean 3-space depending on the design pattern, including the pattern further outside (figure 2b(i)(ii)). It is noteworthy that this kind of pattern enables us to fold the material along a closed curve. Surfaces so moulded are shown in figure 2b. Such folds along curves are possible with our method because of the induced deformations in the metric, in contrast to folds in the style of traditional origami, which are necessarily only along intrinsic straight lines (geodesics) on a sheet of paper, as the intrinsic metric remains unchanged in origami. (There are modifications to origami designed to overcome this restriction [30].)

(viii) The thickness of the material that we presently use makes it difficult to go below sizes smaller than a few millimetre (figure 7d,e) but this is not a fundamental limitation. Indeed, thinner thermo-responsive materials could be used after solving the problem of how to deposit the needed heat-responsive patterns. However, the problem of undesired thermal conduction is likely to become more acute as the size becomes smaller.

(ix) It follows from the electronic supplementary material, figure S3 that the stresses exerted by the contraction of the black regions are greater than 0.5 MPa. This stress is sufficient to bend the white regions. Once the designed contraction has taken place the heating is switched off. This leads to a rapid rise in the elasticity (from the order of 1 MPa to the order of 1 GPa) of the black region which had turned soft during the heating. With the restoration of the elasticity, the stresses created by the altered geometry hold the various regions in their new bent shapes.

## 3.2. Additional remarks on practical implementation

It is desirable to transfer heat very rapidly (flash heating), which has the twin advantages that the change of shape, which happens more slowly compared with the time scale of rapid heating, does not interfere with the scheme of heating by radiation and the white (non-radiated) region remain cold, which would otherwise have heated up by conduction during a longer process of heating by lower intensity radiation. One should note that a curved object with a different global topology than that of a flat sheet will have to be made by cutting and gluing together individual curved pieces moulded by the above method. The reader's attention is invited to the electronic supplementary material where the degree to which we succeed in getting the desired shapes is quantified for some examples.

# 4. One-dimensional moulding

Before we come to moulding surfaces, it is useful to consider a simplified one dimensional version of the problem. Suppose that we wish to convert a one-dimensional strip of length $L_0$ into a strip of length $L_1$ after contracting an appropriately chosen part of it by a constant coefficient $\gamma$ ($0 < \gamma < 1$), where we must assume that $\gamma L_0 \leq L_1 \leq L_0$. If $L_0$ is made up of a white portion of length $w$ which does not contract and a black portion of length $b$ that contracts by the coefficient $\gamma$, then we get the system of simultaneous equations $w + b = L_0$, $w + \gamma b = L_1$. Solving this gives the unique solution

$$w = \frac{L_1 - \gamma L_0}{1 - \gamma} \quad \text{and} \quad b = \frac{L_0 - L_1}{1 - \gamma}. \tag{4.1}$$

In a one-dimensional moulding problem, we can divide $L_0$ into any sequence of white and black segments such that the total white length is $w$ and total black length is $b$, and then heat it to get the length $L_1$. The actual arrangement of these segments does not matter.

## 4.1. The case of a periodic one-dimensional moulding (figure 3)

We include here the following one-dimensional calculation which will be important for later use in §§6 and 7. Suppose that we want the one-dimensional black and white pattern along a long strip to be

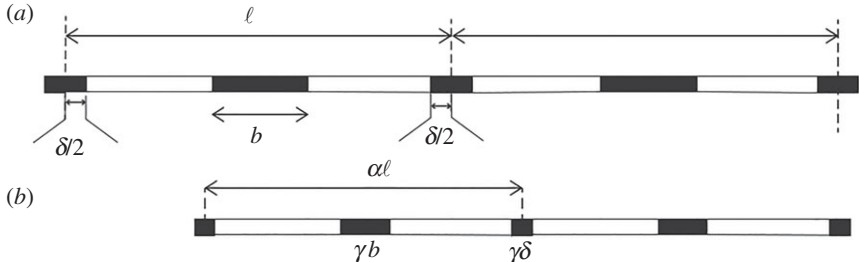

**Figure 3.** The figure schematically shows a one-dimensional periodic pattern of black and white patches (*a*) and the result after heating (*b*). The black patches contract in length by the multiplier $\gamma$ while the white patches retain their length.

periodic with period $\ell$. This means there will exist a real number $\alpha$ with $\gamma \leq \alpha \leq 1$ such that each segment of length $\ell$ will contract to give a segment of length $\alpha\ell$ after moulding, and the result after moulding will be periodic with period $\alpha\ell$. The value $\alpha = \gamma$ corresponds to an entirely black pattern, and the value $\alpha = 1$ corresponds to an entirely white pattern. Suppose that the pattern in a basic segment $[0, \ell]$ is as follows. There are numbers $\delta$ and $b$ such that

$$0 < \frac{\delta}{2} < \frac{\ell - b}{2} < \frac{\ell + b}{2} < \ell - \frac{\delta}{2} < \ell, \tag{4.2}$$

and the three black segments are $[0, \delta/2]$, $[(\ell - b)/2, (\ell + b)/2]$, $[\ell - \delta/2, \ell]$, and the remaining two segments $[\delta/2, (\ell - b)/2]$ and $[(\ell + b)/2, \ell - \delta/2]$ are white. The value of $\delta$ is the smallest width of a black portion for which the heating is effective, without too much loss by conduction, which is approximately 4 mm for our experimental set-up. We want the length of the middle black segment to be $\geq \delta$, which means

$$b \geq \delta. \tag{4.3}$$

Also, $\ell \geq b + \delta$, so we have

$$b \leq \ell - \delta. \tag{4.4}$$

The end black portion of length $\delta/2$ in one basic segment is contiguous with the beginning black portion of length $\delta/2$ in the next basic segment, so together they have a contractible length $\delta$. As the total black portion in the basic segment has length $b + \delta$, which contracts to $\gamma(b + \delta)$ on heating, the basic segment contracts to a new length $\ell - b - \delta + \gamma(b + \delta) = \ell - (1 - \gamma)b - (1 - \gamma)\delta$. Hence we must have

$$\frac{\ell - (1 - \gamma)b - (1 - \gamma)\delta}{\ell} = \alpha, \tag{4.5}$$

which on solving for $b$ gives

$$b = \left(\frac{1 - \alpha}{1 - \gamma}\right)\ell - \delta. \tag{4.6}$$

.

As we must have $\delta \leq b \leq \ell - \delta$, this gives

$$\gamma \leq \alpha \leq 1 - 2(1 - \gamma)\frac{\delta}{\ell}. \tag{4.7}$$

We can change the original problem by changing $\alpha$ to $\alpha/c$, which is the result of a $c$-trick. Hence if the original $\alpha$ does not satisfy the above inequalities, we choose a $c$ such that $\alpha/c$ satisfies them, that is, we must choose a value of $c$ such that

$$c \in \left[\frac{\alpha}{1 - 2(1 - \gamma)\frac{\delta}{\ell}}, \frac{\alpha}{\gamma}\right]. \tag{4.8}$$

Such a value of $c$ exists as the above interval is non-empty, which follows from inequality (4.6).

In the case where we want $\alpha$ to vary from one lattice segment $[n\ell, (n+1)\ell]$ to other, then $b$ will vary across these lattice segments. In order that a common constant $c$ exists, we must have

$$\frac{\alpha_{\max}}{1 - 2(1-\gamma)\dfrac{\delta}{\ell}} \leq \frac{\alpha_{\min}}{\gamma}. \tag{4.9}$$

This can be satisfied by taking $\delta/\ell$ to be sufficiently small provided we have

$$\frac{\alpha_{\min}}{\alpha_{\max}} \geq \gamma. \tag{4.10}$$

The above inequality needs to be satisfied by our moulding problem for the given physical material which has contraction coefficient $\gamma$.

# 5. Contraction tailoring

The standard tailoring method to produce an approximately curved surface from a cloth is to cut out and discard curvilinear wedges (darts) from the cloth and then to stitch together two of the resulting edges [31]. Sometimes, a piece shaped like an eye is cut out, and the two edges are stitched together. Instead of cutting out the wedges, one can sometimes form folds in the style of origami [32,33], or 'pleats' as in many common garments [31], to achieve a somewhat similar result, but with the presence of folds.

It is to be noted that tailoring does not affect the Gaussian curvature[1] $\kappa$ of the cloth away from the stitches, where it remains flat (means $\kappa$ remains 0). In a tailored garment, the curvature is concentrated near the stitches, where the material deforms a bit, and also there are singularities such as vertices of cones and edges of pleats, where the intrinsic or extrinsic[2] curvatures get concentrated. The process of cutting out darts and bringing two edges close can be approximated by the contraction produced by selective heating of a pattern of darts. The stiffness of plastic (in contrast to the floppiness of cloth) allows us to use tailoring to fashion a shape in $\mathbb{R}^3$, i.e. to have a prescribed embedding in $\mathbb{R}^3$ up to isometries of $\mathbb{R}^3$.

Unlike the other methods (metric moulding and distance moulding) that we discuss later, in which we specify an algorithm to achieve a given shape, we do not suggest a general algorithm for contraction tailoring, except in the case of suitable surfaces of revolution.

## 5.1. Surfaces of revolution

Let $r = (x^2 + y^2)^{1/2}$ denote the radial distance from the $z$-axis in $\mathbb{R}^3$. Suppose that a surface $M \subset \mathbb{R}^3$ is a surface of revolution around the $z$-axis, which is topologically either a disc or an annulus. In parametric terms, such an $M$ can be given as follows. In the case where $M$ is topologically a disc, it must intersect the $z$-axis in a single point $(0, 0, z_0)$. In the case where $M$ is topologically an annulus, the inner perimeter of the annulus will correspond to a circle $z = z_0$, $(x^2 + y^2)^{1/2} = s_0$ on $M$ of radius $s_0 > 0$. The family of planes $y\cos\theta - x\sin\theta = 0$ in $\mathbb{R}^3$, parametrized by the angle $\theta$, will intersect $M$ in a family of geodesics $C_\theta$. Let $s$ denote the arc-length along any such geodesic, measured by starting with the initial value $s = s_0$. In the case where $M$ is homeomorphic to a disc, the inner perimeter of the annulus is just a point, and we have $s_0 = 0$. The surface $M$ is parametrically given by $x = r(s)\cos\theta$, $y = r(s)\sin\theta$, and $z = h(s)$, where $r(s)$ and $h(s)$ are functions of $s$. Let $s$ vary from the starting value $s_0$ to a maximum value $s_1$. We assume that the functions $r, h : [s_0, s_1] \to \mathbb{R}$ are sufficiently smooth. As $s$ is the arc-length along the radial geodesics on $M$ we have $ds^2 = dr^2 + dz^2$, hence

$$\left(\frac{dr}{ds}\right)^2 + \left(\frac{dh}{ds}\right)^2 = 1, \tag{5.1}$$

which gives us the inequality

$$\left|\frac{dr}{ds}\right| \leq 1. \tag{5.2}$$

---

[1]Recall that the Gaussian (or intrinsic) curvature $\kappa$ is zero, positive or negative at a point, if the ratio of circumference to radius of a small circle on the surface centred at that point is equal to, less than or greater than $2\pi$, respectively.

[2]The extrinsic curvature is captured by the second fundamental form. Its eigenvalues $\kappa_1$ and $\kappa_2$ are the principal curvatures at a point, and their product equals the Gaussian curvature, which is the intrinsic curvature $\kappa$. It is intrinsic in the sense that it depends only on the induced Riemannian metric on the surface, and not directly on its embedding into $\mathbb{R}^3$.

Note that we must have $r(s) > 0$ for all $s_0 < s \leq s_1$, and $r(s_0)$ is 0 or strictly positive depending on respectively whether $M$ is homeomorphic to a disc or an annulus. If $s_0 = 0$, then the corresponding point $(0, 0, z_0)$ on $M$ (which is where $M$ intersects the z-axis) is a singular point on $M$ unless $dh/ds = 0$ at $s = 0$.

Let $D \subset \mathbb{R}^2$ be the annulus centred at the origin with inner radius $s_0$ and outer radius $s_1$, which is a disc in the case where $s_0 = 0$. Let $s$ denote the distance from the origin, and $\theta$ the angle, so that $D$ has polar coordinates $(s, \theta)$. We define $\varphi : D \to M$ by $(s, \theta) \mapsto (r(s) \cos \theta, r(s) \sin \theta, h(s))$. This is a homeomorphism, which takes the radii of $D$ isometrically to the geodesics $C_\theta$ on $M$. The circle $\Gamma_s$ of radius $s$ on $D$ centred at the origin, which has perimeter $2\pi s$, goes to the circle defined on $M$ by the two equations

$$r = r(s) \quad \text{and} \quad z = h(s), \tag{5.3}$$

whose perimeter is $2\pi r(s)$. As $r(s_0) = s_0$, and as $|dr/ds| \leq 1$, we must have $2\pi r(s) \leq 2\pi s$. Hence each circle $\Gamma_s$ contracts under the map $\varphi$ to give the circle $\varphi(\Gamma_s)$ on $M$. In fact, unless $h$ is a constant function on $[s_0, s]$, we will have a strict inequality $2\pi r(s) < 2\pi s$.

Being a surface of revolution, the Gaussian curvature of $M$ is a function of $s$ alone, given by the formula

$$\kappa(s) = \frac{h'(s)h''(s)r'(s) - h'(s)^2 r''(s)}{r(s)(h'(s)^2 + r'(s)^2)^2}. \tag{5.4}$$

Based on the function $r(s)$ on $D$ (but without using the function $h(s)$), we now make a pattern of black wedge-like shapes on $D$. The map $\varphi : D \to M$ keeps the radial distances in $D$ constant and reduces the circumferential length by contraction in the angular direction by the factor $r(s)/s$. The circumferential reduction can be achieved by drawing a suitable wedge-shaped pattern. The total breadth $b(s)$ of all the black wedges intersected with the circle $\Gamma_s$ is then given in terms of equation (4.1) by

$$b(s) = \frac{2\pi s - 2\pi r(s)}{1 - \gamma}. \tag{5.5}$$

This approach is schematically shown in figure 4. Examples of surfaces of revolution that are fabricated following this method is shown in figure 5.

However, it is important to notice that a surface $M$ involves two functions $r(s)$ and $h(s)$, but our recipe for tailoring it by contraction just uses the single function $r(s)$, and so it is susceptible to the following ambiguity as there is no direct control on $h(s)$. Consider two functions $h_1, h_2 : [s_0, s_1] \to \mathbb{R}$ such that there is a point $s^* \in (s_0, s_1)$ with the following properties:

(i) $h_1(s) = h_2(s)$ for $s < s^*$,
(ii) $h_1(s^*) = h_2(s^*)$,
(iii) $h_1(s) + h_2(s) = 2 h(s^*)$ for $s > s^*$ and
(iv) $(dh_1/ds)(s^*) = 0$, $(dh_1/ds)(s) < 0$ for $s < s^*$ and $(dh_1/ds)(s) > 0$ for $s > s^*$.

Consequently, $(dh_2/ds)(s^*) = 0$, $(dh_2/ds)(s) < 0$ for $s < s^*$ and $(dh_2/ds)(s) < 0$ for $s > s^*$. Let surfaces $M_1$ and $M_2$ be defined, respectively, by the pairs of functions $(r(s), h_1(s))$ and $(r(s), h_2(s))$ where $r(s)$ is common. Notice that if $(dr/ds)^2 + (dh_1/ds)^2 = 1$, then automatically $(dr/ds)^2 + (dh_2/ds)^2 = 1$ as $dh_1/ds = \pm dh_2/ds$. These two surfaces coincide for $s \leq s^*$, but are reflections of each other in the plane $z = h_1(s^*)$ for $s \geq s^*$. As $r(s)$ is common for $M_1$ and $M_2$, the thickness function $b(s)$ for black wedges is the same for both these surfaces. This raises the question of how to selectively get $M_1$ or $M_2$ by moulding the flat sheet $D$. Also note that the Gaussian curvature for $M_1$ and $M_2$ is given by the *same* function $\kappa(s)$, as both $h'$ and $h''$ change sign in formula (5.4) for $\kappa(s)$.

We can resolve this ambiguity and produce $M_1$ or $M_2$ selectively as desired, using the following fortunate circumstance which was discussed in §3.1.1(iii). When portions of $D$ are painted black from one side of $D$ and heated, the temperature rises more on the side which is painted which makes that side contract more, and so $D$ has a propensity to bend—much as a bi-metallic strip—towards the hotter side. Hence to make $M_1$, the piece $D$ will be painted on one side only, while to make $M_2$, the painting is on opposite sides for $s < s^*$ and $s > s^*$ (figure 6).

To avoid problems associated with conduction of heat between neighbouring areas, the width of any black wedge should not be too small. On the other hand, if the width of a black region is too large, then some undesired instabilities can result in buckling and contortions. In order to keep the widths of the black wedges in an effective range, which is about 4–6 mm, the number $n(s)$ of wedges can be varied with $s$, so that $b(s)/n(s)$ lies in this effective range.

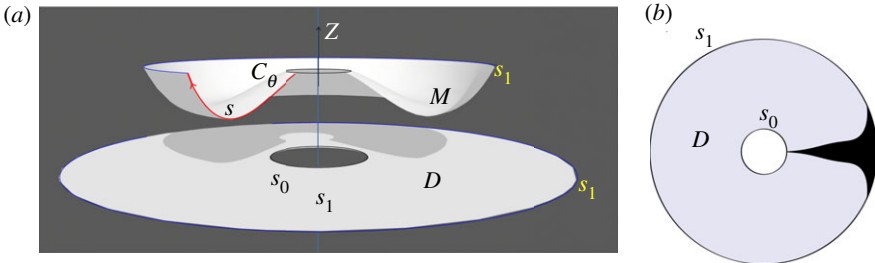

**Figure 4.** Contraction-tailoring method. The panel (*a*) shows a surface of revolution *M*, and a plane annulus *D* from which it is to be fashioned. Instead of showing a number of black bands with total width *b*(*s*), the panel (*b*) shows for simplicity a single black band drawn on *D* whose width is equal to the required total width *b*(*s*) of the black bands.

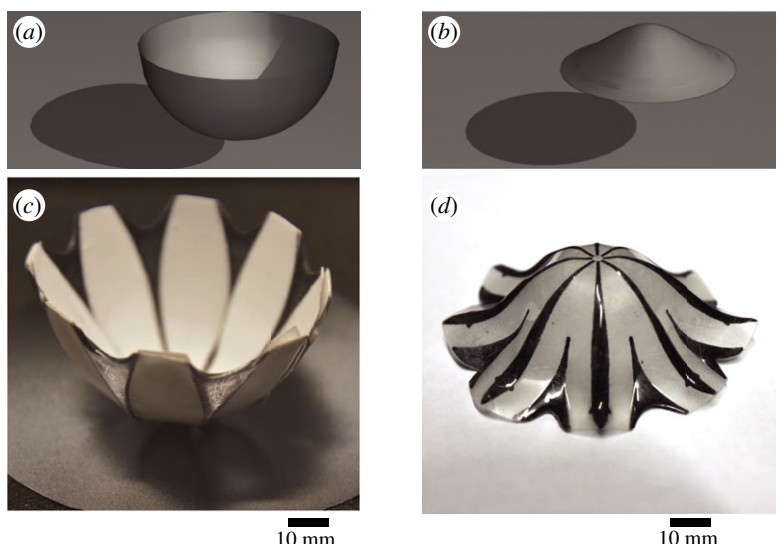

**Figure 5.** Panels (*a*,*b*) show the mathematical examples of surfaces of revolution. Panels (*c*,*d*) show the experimental results obtained by contraction tailoring using the method prescribed in the text.

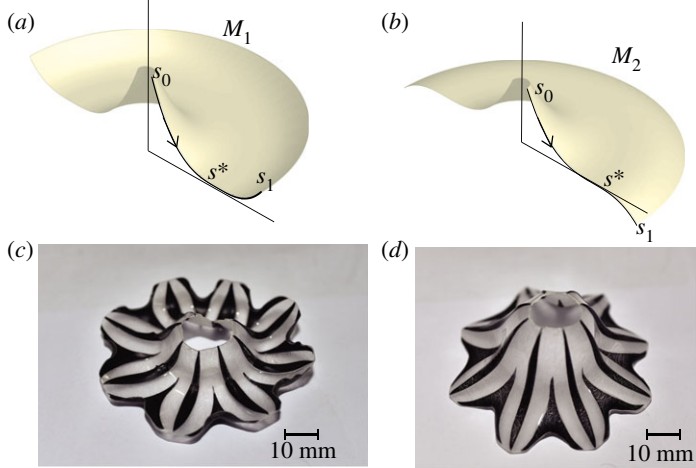

**Figure 6.** Resolution of ambiguity of embedding. As explained in the text, under certain conditions two different surfaces of revolutions, such as the surfaces $M_1$ and $M_2$ shown in panels (*a*) and (*b*), correspond to the same function *r*(*s*). However, we can mould $M_2$ by changing the side of *D* that is painted starting from the critical point $s = s^*$ of *h*(*s*), while if we paint *D* always on the same side then it will result in $M_1$. The photographs of the physical realizations of $M_1$ and $M_2$ are shown in panels (*c*) and (*d*), respectively.

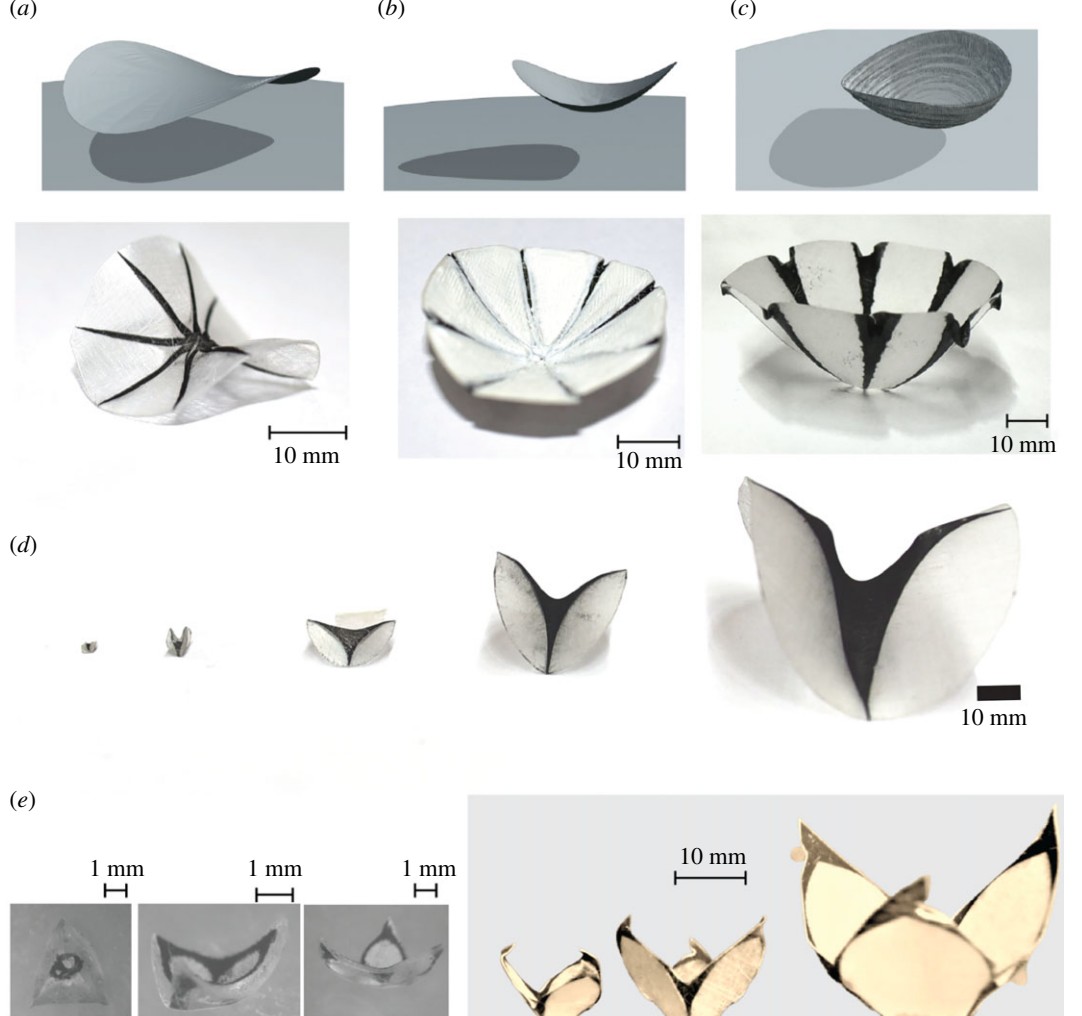

**Figure 7.** Panels (*a–c*) show various surfaces which are *not* surfaces of revolution, that were fashioned by the contraction-tailoring method. Panels (*d,e*) show a sequence of similar moulded shapes of different sizes.

## 5.2. The algorithmic procedure for moulding a surface

The algorithmic procedure for moulding a surface of revolution $M$ has the following steps.

(i) Numerically specify the defining functions $r(s)$ and $h(s)$ on a specified domain $[s_0, s_1]$. These should be sufficiently smooth and have the following properties: (a) $r(s_0) = s_0$, and $r(s) > 0$ if $s > s_0$, (b) $(dr/ds)^2 + (dh/ds)^2 = 1$.

(ii) Take a piece $D$ of plastic, which is an annulus of inner and outer radii $s_0$ and $s_1$, respectively. In the special case $s_0 = 0$, $D$ is a disc of radius $s_1$.

(iii) Calculate the function $b(s) = 2\pi(s - r(s))/(1 - \gamma)$ where $\gamma$ is the contraction coefficient of the plastic material of $D$.

(iv) Draw radial black wedges, whose number $n(s)$ depends on $b(s)$ by the requirement that $\alpha \leq b(s)/n(s) \leq \beta$, where $\alpha$ and $\beta$ are the chosen minimum and maximum widths, respectively. The wedges are spread uniformly along the angular parameter $\theta$, and their total width is $b(s)$.

(v) Heat the piece $D$ by infrared radiation.

**Remark 5.1.** It is possible to mould many interesting surfaces by contraction tailoring which are not surfaces of revolution, though we do not have a general algorithm for doing so. Figure 7*a–c* shows some examples of these.

**Remark 5.2.** If homogeneity of the infrared illumination is maintained over large areas, this method of moulding can be scaled and applied from a few millimetre upwards, simply by adhering to the design

requirement that individual black or white regions should not be too large or too small. That is, if we want to make much larger objects then instead of just scaling up the inset designs as in figure 7*d*,*e*, we will have to further break up the black regions and spread these among the white regions, so that the individual black or white regions do not become too large. The thickness of the material that we presently use makes it difficult to go below sizes smaller than a few millimetres, but this is not fundamental. Indeed, thinner thermo-responsive materials [34] could be used after solving the problem of how to deposit the needed heat-responsive patterns.

# 6. Metric moulding

In this section, we describe a method of moulding which is geared towards altering the original Euclidean metric on the plastic sheet so that we get the desired new Riemannian metric by selective contractions. It is possible to convert this method into an algorithmic procedure. The desired new metric is not required to have any special symmetry (e.g. rotational symmetry).

We begin with a chosen diffeomeorphism $\varphi : D \to M$, which we can assume is everywhere a local contraction (by the *c*-trick as explained in §2). The Riemannian metric on $M$ is induced by the Euclidean metric on the ambient $\mathbb{R}^3$. The desired new metric on $D$ is the pullback of the metric on $M$ by $\varphi$. Let $X$, $Y$ be Cartesian coordinates drawn on the flat piece $D$ before it is deformed. The original metric on $D$ prior to deformation is $dS^2 = dX^2 + dY^2$. The desired new metric on $D$ therefore has the form $ds^2 = E\,dX^2 + 2F\,dX\,dY + G\,dY^2$, where $E$, $F$, $G$ are functions of $X$, $Y$ with $E > 0$, $G > 0$ and $EG - F^2 > 0$. The functions $E$, $F$ and $G$ are given in terms of $\varphi$ by equation (2.2).

Note that at any point $P$ of $D$, the $2 \times 2$-matrix $\begin{pmatrix} E(P) & F(P) \\ F(P) & G(P) \end{pmatrix}$ is symmetric positive definite, so there exists an orthonormal frame $u(P)$, $v(P)$ w.r.t. the flat metric $dS^2$ at the point $P$ which diagonalizes the above matrix, so that $u(P)$ and $v(P)$ are eigenvectors with eigenvalues $0 < \lambda(P)$, $\mu(P)$. In the special case $\lambda(P) = \mu(P)$, any pair of orthogonal vectors can serve as $u(P)$, $v(P)$. In a small enough neighbourhood of any point, we can treat $u$, $v$, $\lambda$ and $\mu$ as continuous single-valued functions of $X$, $Y$. As $\varphi$ was chosen to be everywhere a local contraction, we must have $\lambda, \mu \leq 1$.

Under the deformation of the flat sheet into the curved surface, a tiny square of side $\ell$ on $D$ will turn approximately into a parallelogram (which will be a rectangle in the special case when the sides of the square are parallel to the eigenvectors). Our moulding strategy is to divide $D$ into a lattice of small squares, approximate the continuous functions $\lambda$, $\mu$, $u$, $v$ by piecewise constant functions that are constant in each square, and paint each of these squares appropriately so that the resulting contraction will change them into the corresponding small parallelograms. The idea is to make these parallelograms fit together to give an approximation of the Riemannian metric of $M$.

The above idea has a problem coming from the following two mismatches: (i) the common edge between two lattice squares gets two different contraction coefficients from the two squares as each must turn into a parallelogram of different dimensions, and (ii) the total angle around a vertex which is $2\pi$ to begin with now becomes the sum of the corresponding angles of the four surrounding parallelograms, which may not add to $2\pi$. This produces tensions which are resolved by an interpolation if the region near the edges and vertices of the lattice squares becomes soft while moulding. We induce such a softening by having a band of a fixed width $\delta/2$ all along the boundary within each lattice square. The value of $\delta$ is the minimum width for which a black patch contracts. On the square lattice, this means that the horizontal and the vertical lattice lines are narrow black bands of width $\delta$. The large-scale effect of these bands is a constant isotropic contraction.

## 6.1. The special case where the desired new metric tensor is constant on D

If the desired new metric tensor $g$ is constant on $D$, then there exists an angle $\theta$ with $0 \leq \theta \leq \pi/2$ such that the basis $u = e_1\cos\theta + e_2\sin\theta$ and $v = -e_1\sin\theta + e_2\cos\theta$ diagonalizes the new metric, with eigenvalues $\lambda$ and $\mu$. This means that in the new metric, $u$ and $v$ remain perpendicular, with new lengths $\|u\|_g = \sqrt{\lambda}$, $\|v\|_g = \sqrt{\mu}$. We assume that $0 < \lambda, \mu < 1$, as we desire that that change is everywhere a contraction. Thus, to bring about the metric $g$, we need to contract $D$ in the direction $u$ by the multiplier $\sqrt{\lambda}$, and contract $D$ in the direction $v$ by the multiplier $\sqrt{\mu}$. Given $\theta, \lambda, \mu$, the corresponding $g$ is given by

$$\begin{pmatrix} E & F \\ F & G \end{pmatrix} = \begin{pmatrix} \lambda\cos^2\theta + \mu\sin^2\theta & (\lambda - \mu)\cos\theta\sin\theta \\ (\lambda - \mu)\cos\theta\sin\theta & \mu\cos^2\theta + \lambda\sin^2\theta \end{pmatrix}. \tag{6.1}$$

A simple but basic example of a map $\varphi : \mathbb{R}^2 \to \mathbb{R}^3$ for which the pullback $g$ of the Euclidean metric is constant is when $\varphi$ is an injective linear map followed by a translation. By choosing new Cartesian coordinates on $\mathbb{R}^3$, we just have to consider the case when $\varphi$ is an invertible linear map $T : \mathbb{R}^2 \to \mathbb{R}^2$. Then as a $2 \times 2$-matrix, we have $g = {}^t T T$ where ${}^t T$ denotes the transpose of $T$. If $T = UA$ is the polar decomposition of $T$, where $A$ is a positive definite symmetric matrix and $U$ is an orthogonal matrix with $\det(U) = 1$, then $g = {}^t A \, {}^t U \, UA = A^2$, where we have used the equalities ${}^t A = A$ and ${}^t U = U^{-1}$. Hence, the eigenvalues of $g$ are exactly the squares of the eigenvalues of the symmetric part $A$ in the polar decomposition of $T$. The orthogonal part $U$ of the polar decomposition physically refers to how the moulded piece is placed in $\mathbb{R}^2$, while the symmetric part $A$ tells us what happens internally to $D$ in the process of moulding. The matrix $A$ has the two mutually perpendicular non-zero eigenvectors $u, v$, with eigenvalues $\sqrt{\lambda}, \sqrt{\mu}$, so $A^2$ has these same eigenvectors with eigenvalues $\lambda, \mu$. The internal modification of $D$ corresponds to linear multiplications by the factors $\sqrt{\lambda}, \sqrt{\mu}$ in the two mutually perpendicular directions $u, v$. It should be noted that we need a polar decomposition of $T$ in order to get these directions $u$ and $v$ and the contraction factors $\sqrt{\lambda}$ and $\sqrt{\mu}$. Once again, we will only allow those $T$ for which $0 < \lambda, \mu < 1$.

Let $\Lambda \subset \mathbb{R}^2$ be a lattice in $\mathbb{R}^2$ (means a discrete subgroup which spans $\mathbb{R}^2$). Suppose $D$ is a large piece in $\mathbb{R}^2$, and suppose we give it a black and white pattern that is periodic w.r.t. $\Lambda$. Then on heating, $D$ will become a plane piece up to small local wiggles which are periodic w.r.t. a new lattice $\Lambda'$. There will be a linear transformation $T : \mathbb{R}^2 \to \mathbb{R}^2$ such that $\Lambda' = T\Lambda$, and the modification in $D$ (up to small periodic wiggles) is given by $T$. If the scale of $\Lambda$ is very small compared with the size of $D$, then we can regard the resulting metric (after moulding) as the constant metric $g = {}^t T T$.

We now choose the lattice $\Lambda \in \mathbb{R}^2$ to be the square lattice of sides $\ell$, with lattice points $(m\ell, n\ell)$ where $m, n$ are integers. The basic two-dimensional problem is that given $\lambda, \mu, \theta$, how to find a black and white pattern with periodicity $\Lambda$, such that on heating the resulting contraction is described (in the large) by a linear transformation $T$ which corresponds to the given $\lambda, \mu, \theta$. Moreover, the pattern should be such that the outer boundary of each lattice square is black of width $\delta/2$ (which is, as explained above, essential for interpolations when $\lambda, \mu, \theta$ vary from lattice square to lattice square).

Suppose that we have a large piece $D$ of plastic in the $x, y$-plane across which we have a black band $B$. The sides of the band are straight lines, parallel to each other. The length of the band is much larger than its width. On heating, the width of the band will shrink by the factor $\gamma$, pulling together the white parts on either side, as happens in plate tectonics. The sides of the band being anchored in large white regions, they cannot shrink. However, there will be a shrinking effect at the ends of the black band, which will result in the these ends getting pulled inwards. Let the band $B$ make an angle $\theta$ with the $x$-axis. The shrinkage of the band is in the perpendicular direction to the band, that is, the angular direction $\theta \pm \pi/2$. If $b$ is the width of the band (means the length of the intersection of the band with a line making the angle $\theta + \pi/2$ with the $x$-axis), then after shrinking the width becomes $\gamma b$.

Next suppose we have two mutually perpendicular black bands $B_1$ and $B_2$ on the plastic, as shown in figure 8a. Let these make angles $\theta$ and $\theta \pm \pi/2$ with the $x$-axis. On heating, the widths $b_1$ and $b_2$ of both the bands get multiplied by $\gamma$. As a result, if we have an imaginary square $R$ of size $L \times L$ on the plastic whose sides are parallel to $B_1$ and $B_2$, through which both these bands pass, then it gets converted into a rectangle whose sides are parallel to the original sides, but now have the modified lengths $L - (1 - \gamma)b_1$ and $L - (1 - \gamma)b_2$ (the original square and the modified rectangle are shown in green in figure 8a and b, respectively). Next, suppose that the plastic is drawn with a criss-cross doubly periodic pattern of mutually perpendicular bands making angles $\theta$ and $\theta + \pi/2$ with the $x$-axis, so that in any large square of size $L$ with sides parallel and perpendicular to the bands, the total width of the bands making angle $\theta$ with the $x$-axis is $b_1$ and the total width of bands making angle $\theta + \pi/2$ with the $x$-axis is $b_2$. (We do not have to assume here that the horizontal period is equal to the vertical periods in this doubly periodic pattern.) Then on heating, such an $L \times L$ square gets converted into a rectangle whose sides are parallel to the original sides, but now have the modified widths $L - (1 - \gamma)b_1$ and $L - (1 - \gamma)b_2$. Thus, on a large scale (up to local variations), the effect of the shrinkage is to convert the original metric $ds^2 = dx^2 + dy^2$ on the plastic into a new metric $E\,dx^2 + 2F\,dx\,dy + G\,dy^2$, where $E = \lambda\cos^2\theta + \mu\sin^2\theta$, $F = (\lambda - \mu)\cos\theta\sin\theta$ and $G = \lambda\sin^2\theta + \mu\cos^2\theta$, where

$$\lambda = \left(1 - (1 - \gamma)\frac{b_1}{\ell}\right)^2 \quad \text{and} \quad \mu = \left(1 - (1 - \gamma)\frac{b_2}{\ell}\right)^2. \tag{6.2}$$

This metric has eigenvectors $e_1\sin\theta - e_2\cos\theta$ and $e_1\cos\theta + e_2\sin\theta$, with eigenvalues $\lambda$ and $\mu$, respectively. In particular, if $b_1/L = b_2/L = \beta$, then the outcome is a constant isotropic contraction by the factor $1 - (1 - \gamma)\beta$, an outcome that is independent of the angle $\theta$.

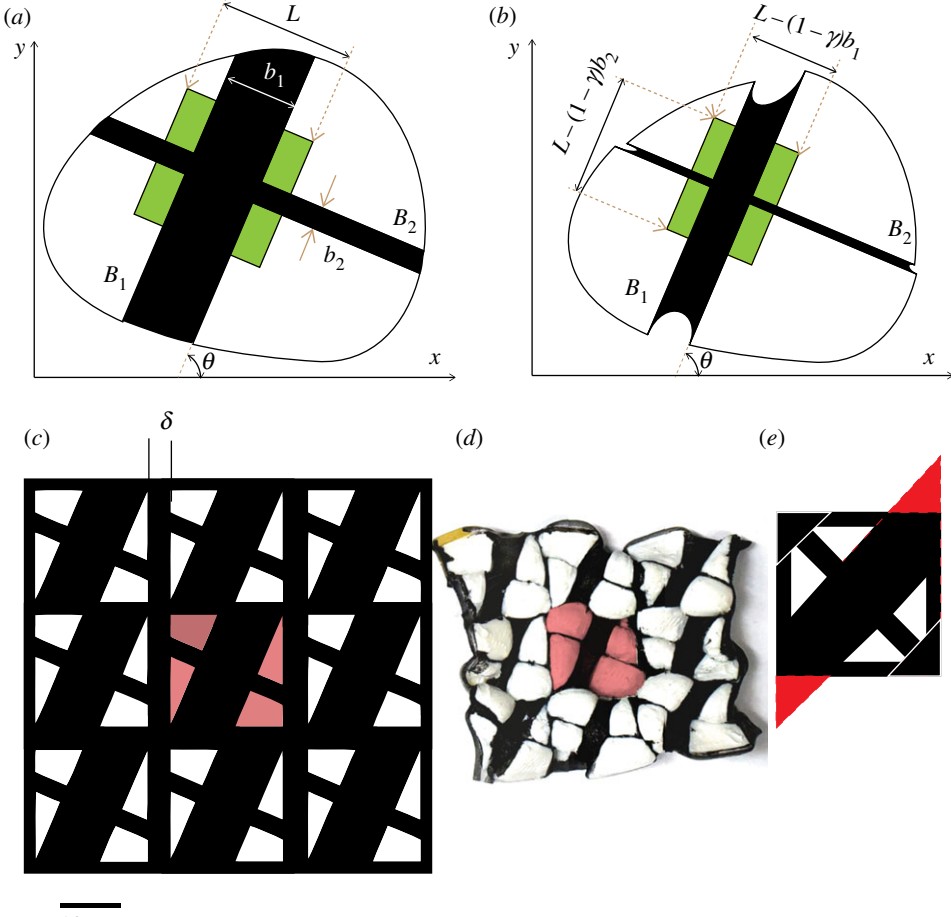

**Figure 8.** Metric moulding. Panels (*a,b*) show the before heating and after heating states of a large piece of plastic, respectively, on which are painted two mutually perpendicular black bands. The widths of these bands shrink by the multiplier $\gamma$, which changes a square with sides parallel to the bands in (*a*) (shaded green) into a rectangle in (*b*). Panel (*c*) shows the periodic tiling pattern which will bring about shrinking by assigned multipliers in two mutually perpendicular directions which make a fixed angle to the basic lattice. Each lattice square has a black border of thickness $\delta/2$. Panel (*d*) shows the rearrangement of the white pieces after contraction. The central patch in (*c*) is depicted in a colour to make it visible how the white patches rearrange themselves when the black portions contract, to give the result shown panel (*d*). The heating also brings about out warping in the white region. When a black band passes through the corner of the lattice square, it has a significant overlap with the black borders. Panel (*e*) shows how to compensate for it by introducing additional black triangles in the remaining two corners.

**Remark.** The individual transformations induced by two mutually perpendicular bands commute with each other. Their order does not matter in a superposition. Also, the uniform isotropic contractions are scalar multiples of identity, so they commute with all transformations. In this way, the basic metric-moulding procedure is non-sequential. This contrasts with general sequential nature of folding in origami where the outcome depends on the order of the folds.

Suppose we have present a superposition of (i) a doubly periodic pattern of mutually perpendicular bands at angles $\theta$ and $\theta + \pi/2$ with respective average densities $\beta_1 = b_1/L$ and $\beta_2 = b_2/L$, and (ii) a doubly periodic pattern of horizontal and vertical bands of equal average densities $\beta_0$, then as the effect of the second pattern is isotropic, one may expect that the combined effect is as if the second pattern is also at angles $\theta$ and $\theta + \pi/2$, and so the combined effect is as if we just have the first pattern modified so that $\beta_1$ and $\beta_2$ are changed to $\beta_0 + \beta_1$ and $\beta_0 + \beta_2$. The problem with this is that there may be a significant overlap between the pattern (i) and the pattern (ii), reducing their effects, as the density of black parts will not simply add up because of the overlaps. While the overlaps between the vertical and horizontal bands within any one pattern is not a problem, non-orthogonal overlaps between two different patterns have to be avoided. As we will see below, our choice of a basic pattern indeed minimizes such non-orthogonal overlaps.

With the above analysis as its heuristic, we now specify our basic pattern for shrinking a flat piece to bring about a new constant metric with given values of $\theta$, $\lambda$, $\mu$, where recall that $\lambda$ and $\mu$ denote the eigenvalues of the metric, and $0 \leq \theta < \pi/2$ is the angle made by an eigenvector $v = e_1\cos\theta + e_2\sin\theta$ corresponding to eigenvalue $\mu$ with the x-axis. Consequently, the eigenvector $u = e_1\sin\theta - e_2\cos\theta$ for $\lambda$ will make the angle $\theta - \pi/2$ with the x-axis.

Figure 8c shows the typical pattern. The pattern is a doubly periodic arrangement of squares, with the same period $\ell$ in the x- and y-directions. A fundamental square in the pattern, which has size $\ell \times \ell$, has as its central feature two mutually perpendicular black bands, which make angles $\theta$ and $\theta + \pi/2$ with the x-axis, where $0 \leq \theta < \pi/2$. These have widths $b_1$ and $b_2$, respectively. Each lattice square has a black border of width $\delta/2$. The value of $\delta$ is chosen to be the minimum width at which thermal contraction becomes effective (so $\delta = 4\,\text{mm}$ in our experiments). The widths $b_1$ and $b_2$ are determined by equation (4.6), taking $\alpha = \sqrt{\lambda}$ and $\alpha = \sqrt{\mu}$, respectively, which gives

$$b_1 = \left(\frac{1 - \sqrt{\lambda}}{1 - \gamma}\right)\ell - 2\delta \quad \text{and} \quad b_2 = \left(\frac{1 - \sqrt{\mu}}{1 - \gamma}\right)\ell - 2\delta. \tag{6.3}$$

By using a sufficiently large value of $c$ in the c-trick, it can be ensured that both $b_1$ and $b_2$ are each greater than $\delta$, so that these black bands contract effectively on heating. Equation (4.8) can be applied taking $\alpha$ to be $\sqrt{\lambda}$ or $\sqrt{\mu}$ to get a range of values of $c$. In order that a common such $c$ exists, by equation (4.9) we must have

$$\frac{\max\{\sqrt{\lambda}, \sqrt{\mu}\}}{1 - 2(1 - \gamma)\dfrac{\delta}{\ell}} \leq \frac{\min\{\sqrt{\lambda}, \sqrt{\mu}\}}{\gamma} \tag{6.4}$$

and then $c$ can be chosen to have any in-between value.

Inequality (6.4) can be satisfied by taking $\delta/\ell$ to be sufficiently small provided we have

$$\frac{\min\{\sqrt{\lambda}, \sqrt{\mu}\}}{\max\{\sqrt{\lambda}, \sqrt{\mu}\}} \geq \gamma. \tag{6.5}$$

The above inequality needs to be satisfied for any $\varphi$ for the given physical material which has contraction coefficient $\gamma$, if the metric-moulding method is to work.

We found that empirical trial and error by varying the widths $b_1$ and $b_2$ of the two central black bands of the pattern can make the moulding more accurate, which gets over the unintended effect of the overlap of the bands $B_1$ and $B_2$ with the black frame of each lattice square.

## 6.2. The general case of a non-constant metric

To produce the colouring pattern to do the desired moulding, we begin by dividing the original flat piece into a square lattice of length $\ell$. As explained above, at the centre $P = (a, b)$ of any lattice square, we have two eigenvectors $u(P)$ and $v(P)$ for the metric tensor $g(P)$ with eigenvalues $0 < \lambda(P), \mu(P) < 1$. We have already given our choice of the periodic pattern (figure 8) which, if drawn in each lattice square, will lead to a uniform contraction corresponding to the data $u(P), v(P), \sqrt{\lambda(P)}, \sqrt{\mu(P)}$. In the general case of a non-constant metric, we draw this pattern only in the lattice square around $P$. Heating this pattern leads to approximately the desired metric on contraction for each square. Note that the adjoining lattice squares have different contraction ratios for the shared edge. Also the sum of the four angles around a vertex may not equal $2\pi$. However, the boundary regions (including the corners) in all the squares are black, and so they become soft on heating, which enables an adjustment which interpolates between the contraction patterns in neighbouring squares along an edge or the four squares around a vertex. If the sum of the angles around the vertex is less than $2\pi$, then the resulting adjustment will produce a region of positive Gaussian curvature around the vertex. Similarly, if the sum of the angles around the vertex is greater than $2\pi$, then the resulting adjustment will produce a region of negative Gaussian curvature around the vertex. In the above, instead of a square lattice, we can use a regular hexagonal lattice, or any other suitable lattice. The choice of what lattice to use also may depend on the approximate symmetry of $M$.

## 6.3. The algorithmic procedure for Riemannian metric moulding

The algorithmic procedure for moulding a surface $M$ which has the desired Riemannian metric has the following steps.

(i) Choose a diffeomorphism $\varphi : D \to M$. One possible method of doing so would be taking the inverse for a vertical projection from $M \subset \mathbb{R}^3$ to the $x$, $y$ plane $\mathbb{R}^2$. This will work in various cases. However, the steps that follow are independent of the choice of $\varphi$.

(ii) Numerically specify the corresponding functions $E$, $F$ and $G$ on $D$.

(iii) Numerically determine the required $c$ factor and replace $D$, $\varphi$ by the corresponding $D^*$, $\varphi^*$. By this device ($c$-trick), we can assume that for the subsequent steps all eigenvalues $\lambda$ and $\mu$ are strictly less than 1. We have to so choose $\ell$ and $c$ such that inequalities (6.4) are satisfied, where the minimum and maximum is now taken over all the lattice squares. It is a necessary condition for this method to work that these inequalities are satisfied.

(iv) Draw the pattern in each lattice square which correspond to the eigenvectors and eigenvalues of the Riemannian metric at the centre of that lattice square.

(v) Heat the piece $D$ by infrared radiation.

If the change of metric is conformal, then $\lambda = \mu$ globally, and the eigenvectors $u$ and $v$ are indeterminate. In such a case, we will take $u = e_1$ and $v = e_2$, which in particular ensures that the intersection of the bands $B_1$ and $B_2$ with the lattice frame is orthogonal. By the Riemann mapping theorem, any metric on a planar region $D$ is conformal to the Euclidean metric on $D$, so one may be tempted to take $\varphi : D \to M$ to be a conformal transformation. However, this is not necessarily practical as the value of $\lambda$ (means the required contraction coefficient) can go outside the achievable range $[\gamma, 1]$. Moreover, the proof of the Riemann mapping theorem does not give a recipe for concretely specifying such a conformal transformation $\varphi$. However, this works well in some examples where the conformal transformation is known and is simple enough, such as the stereographic projection of a domain on a sphere to a planar domain, which then may have to be combined with the $c$-trick which is necessarily conformal.

Instead of using a square lattice, we can use a regular hexagonal lattice in the above procedure, with appropriate hexagonal analogues of the values of the $b_1$ and $b_2$ (in place of equation (6.3)) and with appropriate bounds given by analogues of inequalities (6.4). A hexagon is qualitatively 'more isotropic' than a square, so such a lattice works more uniformly when the direction $\theta$ is changing. The hexagonal design has an additional benefit that (unlike in the case of a square design) the short black segments at the border of any basic hexagon get terminated, instead of prolonging as system-spanning black lines along which unintended folding can occur on heating.

## 6.4. The embedding of $M$ in $\mathbb{R}^3$

A surface embedded in the 3-space is called *rigid* if the only embeddings of it into the 3-space which induce the same Riemannian metric are the rigid translations, rotations and reflections of the original embedding. For example, a sphere (or any dense open subset of it) is rigid. However, open surfaces in general may or may not be rigid, in particular, there exist non-rigid open surfaces with any constant value of Gaussian curvature $\kappa$, positive negative or zero. For example, a hemisphere ($\kappa > 0$) or portions of a cylinder or a cone $\kappa = 0$, or surface similar to that depicted in figures 6 and 7a ($\kappa < 0$) are not rigid.

It follows that when we obtain the Riemannian metric of a rigid surface by deformation of a flat sheet, we automatically obtain its desired shape in $\mathbb{R}^3$ up to translations, rotations and a possible reflection. In particular, when trying to make a chiral object, one may end up with the opposite of the desired chirality.

Given a surface $M \in \mathbb{R}^3$ and a diffeomorphism $\varphi : D \to M$ where $D \subset \mathbb{R}^2$, let $g$ be the pullback to $D$ of the Riemannian metric of $M$ that is induced by its inclusion in $\mathbb{R}^3$. The above metric-moulding method will convert $D$ into a surface $N \subset \mathbb{R}^3$ which has the prescribed intrinsic Riemannian metric $g$, but we may not be able to obtain $M$ from $N$ by a rigid transformation of $\mathbb{R}^3$, as $M$ will not be rigid in general. However, $N$ will have a definite shape in $\mathbb{R}^3$, and this extra structure (beyond its Riemannian metric) comes from the rigidity or elastic properties of mainly the white parts of $D$. Recall that the black parts soften and so easily change their shape during heating, and also, they contract. By contrast, the white material remains stiff throughout, and may undergo only some elastic bending. This raises the question whether we can have another method of moulding $D$, which—instead of trying to get the right Riemannian metric on $M$—directly attempts to get right the embedding of $M$ into $\mathbb{R}^3$, by making use of the enduring stiffness of the white portions of $D$. We present such a method in the following section.

Figure 9 shows the implementation of the above algorithm to mould surfaces of positive and negative curvature. The input pattern that needs to be printed for obtaining these two shapes is given in the electronic supplementary material.

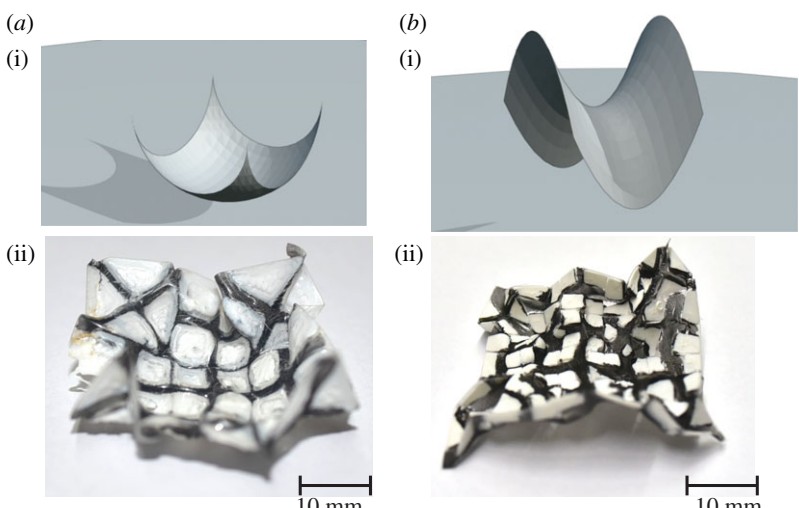

**Figure 9.** Panels (a,b) show the photographs of two surfaces that were made using the metric-moulding method. The shape in (a) is a part of a sphere which has a positive curvature while that in (b) is a part of a saddle that has a negative curvature. The target shapes are given in (a(i),b(i)) while the obtained shapes are in a(ii),b(ii). The corresponding computed input patterns for (a,b) are given in the electronic supplementary material.

# 7. Distance moulding via triangulation

As before, let $M \subset \mathbb{R}^3$ be a surface, and let $\varphi : D \to M$ be a diffeomorphism where $D$ is a domain in $\mathbb{R}^2$. By the c-trick, we can always choose the pair $(D, \varphi)$ in such a way that $\varphi$ is everywhere a contraction. Let $D$ be triangulated (paved) by equilateral triangles as shown in figure 10, and let $T \subset D$ be a basic triangle. Let $\ell$ denote the distance between any vertex $A_i$ of $T$ and the centroid $C_{ijk}$ of $T$. In particular, the basic triangles have sides $\sqrt{3}\ell$. Let $P_1, P_2, P_3 \in M$ be the images under $\varphi$ of the three vertices $A_1, A_2, A_3$ of $T$, let $Q_{ij} \in M$ be the image of the midpoint $B_{ij}$ of the side $A_iA_j$ of $T$, and let $R_{ijk} \in M$ be the image of the centre $C_{ijk}$ of $T$. Let $d_{ij} = \|P_i - Q_{ij}\|$ and $d_{ijk} = \|P_i - R_{ijk}\|$ be the distances in $\mathbb{R}^3$ between these points. The corresponding distances in $T$ are $d(A_i, B_{ij}) = \sqrt{3}\ell/2$ and $d(A_i, C_{ijk}) = \ell$. The distances on $M$ are smaller than the corresponding distances on $D$ because $\varphi$ is a contraction.

The task of moulding is to convert the equilateral triangle $A_iA_jA_k = T$ with sides $\sqrt{3}\ell$ into the curvilinear triangle $T'$ on $M$ which is the image of $T$.

Let $D_{ij}$ be the point on the segment $A_iB_{ij}$ such that

$$\|A_i - D_{ij}\| = \frac{d_{ij} - (\sqrt{3}/2)\ell\gamma}{1 - \gamma}. \tag{7.1}$$

The above distances are so chosen that (see equation (4.1)) if the segment $D_{ij}D_{ji}$ contracts by factor $\gamma$ and the segments $A_iD_{ij}$ and $A_jD_{ji}$ retain their original length, then the original length $\sqrt{3}\ell$ of the segment $A_iA_j$ contracts to become the desired length of the segment $P_iP_j$. Let $E_i$ be the point on the segment $A_iC_{ijk}$ such that

$$\|A_i - E_i\| = \frac{d_{ijk} - \ell\gamma}{1 - \gamma}. \tag{7.2}$$

Once again, these distances are so chosen that if the segment $E_iC_{ijk}$ contracts by the factor $\gamma$, then the original length $\ell$ of the segment $A_iC_{ijk}$ contracts to become the desired length of the segment $P_iR_{ijk}$. The triangle $T$ with these points is shown in figure 10, with a certain polygonal region shaded yellow, which is the region that will be painted black before heating.

Let $\alpha_{ij} = \|P_i - Q_{ij}\|/\|A_i - B_{ij}\| = 2d_{ij}/\sqrt{3}\ell$. As the contraction factor is bounded below by $\gamma$, we must have $\gamma < \alpha_{ij}$. On the other hand, as contraction by heating to be reliably effective, we need to ensure that the relevant width of the yellow region is at least $\delta$. Within the segment $A_iB_{ij}$ which has un-contracted original length $(\sqrt{3}/2)\ell$ the yellow portion $D_{ij}B_{ij}$ is contiguous with the yellow portion $D_{ji}B_{ij}$ of the segment $A_jB_{ij}$ (remember here that $B_{ij} = B_{ji}$), so each of $D_{ij}B_{ij}$ and $D_{ji}B_{ij}$ needs to have length at least $\delta/2$. Hence we must have $\alpha_{ij} < 1 - (\delta/\sqrt{3}\ell)(1 - \gamma)$. Together, we have the bounds

$$\gamma < \alpha_{ij} < 1 - \frac{\delta}{\sqrt{3}\ell}(1 - \gamma). \tag{7.3}$$

This gives a non-empty range for $\alpha_{ij}$ if $\delta/\ell$ is sufficiently small.

(a)

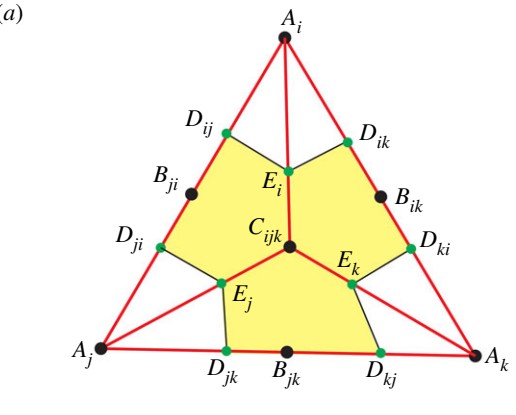

(b)

$\varphi^{-1} = (c \times -) \circ \pi : M \to D$

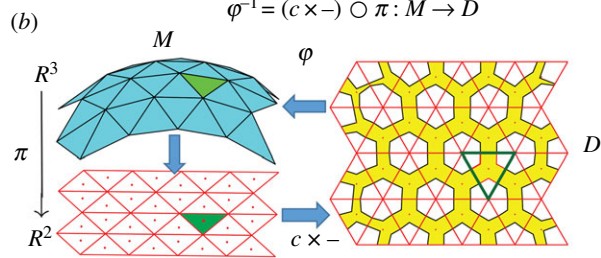

**Figure 10.** (a) The figure shows the shading pattern for a basic triangle $T$ in $D$. (b) This figure shows the example of the distance-moulding pattern for a portion of a sphere. An everywhere contracting $\varphi : D \to M$ is obtained by combining the inverse of the vertical projection $\pi : \mathbb{R}^3 \to \mathbb{R}^2$ and the $c$-trick as explain in §2. The yellow painted region on $D$ is painted black and when heated $D$ moulds into $M$. It is noteworthy how the shading pattern changes as one moves towards the edge of $D$.

Next, let $\alpha_{ijk} = \|P_i - R_{ijk}\| / \|A_i - C_{ijk}\| = d_{ijk}/\ell$. In the segment $A_i C_{ijk}$ which has un-contracted length $\ell$, the yellow portion $C_{ijk}E_i$ is contiguous with the yellow portion $C_{ijk}B_{jk}$ which has length $\ell/2$, which is greater than $\delta/2$. Hence it is enough if $C_{ijk}E_i$ has length $> \delta/2$. Hence we must have $\alpha_{ijk} < 1 - (\delta/2\ell)(1 - \gamma)$. This gives the bounds

$$\gamma < \alpha_{ijk} < 1 - \frac{\delta}{2\ell}(1 - \gamma). \tag{7.4}$$

Again, this gives a non-empty range for $\alpha_{ij}$ if $\delta/\ell$ is sufficiently small.

By replacing the original $\alpha$ by $\alpha/c$ by the $c$-trick, we can ensure that the above simultaneous inequalities hold across all triangles $T$ on $D$ provided that $c$ lies in the range

$$\max\left\{ \frac{\alpha_{ij}}{1 - \frac{(1-\gamma)\delta}{\sqrt{3}\ell}}, \frac{\alpha_{ijk}}{1 - \frac{(1-\gamma)\delta}{2\ell}} \right\} \leq c \leq \min\left\{ \frac{\alpha_{ij}}{\gamma}, \frac{\alpha_{ijk}}{\gamma} \right\} \tag{7.5}$$

where the maximum and minimum are taken over all triangles $T = A_i A_j A_k$ in the triangulation. This shows that we must require that

$$\max\left\{ \frac{\alpha_{ij}}{1 - \frac{(1-\gamma)\delta}{\sqrt{3}\ell}}, \frac{\alpha_{ijk}}{1 - \frac{(1-\gamma)\delta}{2\ell}} \right\} \leq \min\left\{ \frac{\alpha_{ij}}{\gamma}, \frac{\alpha_{ijk}}{\gamma} \right\} \tag{7.6}$$

so that the above range for values of $c$ is non-empty.

Inequality (7.6) can be satisfied by taking $\delta/\ell$ to be sufficiently small provided we have

$$\frac{\min\{\alpha_{ij}, \alpha_{ijk}\}}{\max\{\alpha_{ij}, \alpha_{ijk}\}} \geq \gamma. \tag{7.7}$$

The above inequality needs to be satisfied for any $\varphi$ for the given physical material which has contraction coefficient $\gamma$, if the distance-moulding method is to work. As the above inequalities are

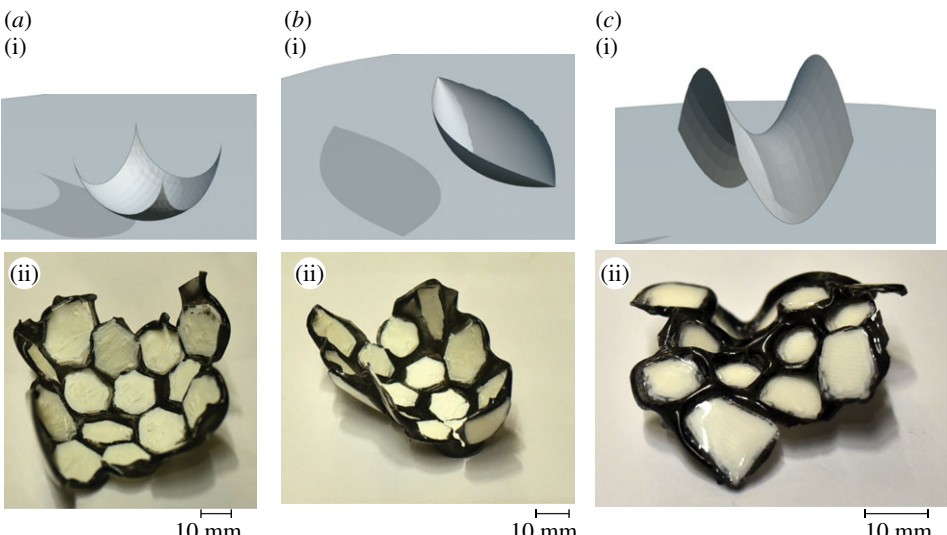

**Figure 11.** Panels (*a–c*) show the photographs of three surfaces that were made using the metric distance method. The target shapes are given in (*a*(i),*b*(i),*c*(i)) while the obtained shapes are in (*a*(ii),*b*(ii),*c*(ii)). The surface in (*a*) is a portion of a sphere. The surface in (*b*) is not a portion of an embedded sphere but it has a constant positive Gaussian curvature, and its Riemannian metric is that of a portion of a sphere. The surface in (*c*) is a portion of a saddle.

satisfied for sufficiently small values of $\gamma$, a more contractable material will enable us to mould a greater range of surfaces.

If a plastic copy of the equilateral triangle $T$, with the yellow region painted black and the rest kept white (or coated with a thick polymer) is heated, then the black region shrinks and consequently the white regions are drawn together. While this happens, the triangle $T$ cannot easily bend by folding along the lines $A_iB_{jk}$ because the presence of the white quadrilateral regions $A_iD_{ij}C_{ijk}D_{ik}$ which remain stiff. The triangle $T$ thereby assumes a new shape which is an approximation of the curvilinear triangle $T' = P_iP_jP_k$ (figure 10), with sides which are approximately of the desired lengths. The middle of the triangle comes out (or goes in: an effect influenced by the bi-metallic strip effect discussed earlier) by approximately the desired extent because of the control of the distances $\|P_iR_{ijk}\|$. The approximation becomes more accurate when instead of a single triangle, we have a lattice of triangles, each of which is given a pattern following the above method. The reasons for this are as follows.

 (i) Adjacent triangles prevent a shrinkage of the black portion $D_{ij}D_{ji}$ of the shared border of the triangles towards the centre of any one of the two triangles, as the adjacent triangle will exert an opposite contracting force, while along the segment $A_iA_j$ the two contractions match.
 (ii) The large number of irregular white regions come in the way of system-spanning long black lines along which unintended folding may occur.

Note that the six white regions around a vertex $A_i$ fit together seamlessly into a polygonal shape with 12 sides, whose 12 vertices are at prescribed distances from the centre $A_i$, which are the same as the distances from $P_i$ to the 12 corresponding points on $M$. When contracted, these polygons get drawn together to the appropriate extent, and the resulting surface approximates the original surface $M$.

Figure 11 shows instances of shapes that were made using the distance moulding via triangulation method. The surface in (*a*) is a part of the sphere. The surface in (*b*) is a part of a spindle which has a constant Gaussian curvature $\kappa$. This surface is parametrically defined by $x(t) = R\cos t$, $y = 0$ and $z(t) = \int_0^t \sqrt{1/\kappa - R^2 \sin^2 \theta}\, d\theta$ for $|t| \leq \pi/2$. Here $R < 1/\sqrt{K}$. The surface in (*c*) is a part of a saddle given by the function $z = x^2 - y^2$. The corresponding computed input patterns for (*a*), (*b*) and (*c*) generated by the algorithm described above are given in the electronic supplementary material. This method of moulding produces satisfactory results at a large enough scale, which is possible when the shape to be moulded is of a size that is much greater than the size of the basic triangle.

# 8. Additional comments

## 8.1. Limitations on moulding

The fact that the constant $\gamma$ is not zero imposes limitations on what can be moulded. For the metric-moulding and distance-moulding methods, the moulding function $\varphi : D \rightarrow M$ needs to satisfy certain inequalities (namely (4.10), (6.5) and (7.7)) which have the generic form

$$\frac{\text{minimum multiplier}}{\text{maximum multiplier}} \geq \gamma, \tag{8.1}$$

which are necessarily satisfied when $\gamma$ is very small. In fact, if $\gamma$ is not 0, there are limitations on what any hypothetical contraction-moulding method can achieve. For example, suppose that we want to mould a portion of the sphere $S_R^2$ of radius $R$ defined in $\mathbb{R}^3$ by the equation $x^2 + y^2 + z^2 = R^2$. If $\gamma = 0$, then it is possible to mould the surface $M = S_R^2 - \{P\}$, which is the complement of a single point (say the north pole $P$) on the sphere $S_R^2$. Now suppose $\gamma > 0$, and we apply the tailoring method to mould $M$, which is the surface of revolution $x^2 + y^2 + z^2 = R^2$, $z \neq R$. The map $\phi : D \rightarrow M$ produced by the tailoring method will begin with a $D$ a disc. Let $O \in D$ denote the centre of $D$, and let $(s, \theta)$ denote polar coordinates on $D$. The map $\varphi$ sends $(s, \theta) \in D$ to the point

$$\left( R \sin \left( \frac{s}{R} \right) \cos \theta, \ R \sin \left( \frac{s}{R} \right) \sin \theta, \ -R \cos \left( \frac{s}{R} \right) \right) \in M. \tag{8.2}$$

The above formula for $\varphi$ shows that the circle with centre $O$ and radius $s$ in $D$ maps to a circle of radius $r(s) = R \sin (s/R)$ in $M \subset \mathbb{R}^3$. This contracts its circumference by the factor

$$\frac{r(s)}{s} = \frac{\sin (s/R)}{s/R}. \tag{8.3}$$

For such a contraction to be practically possible with the given physical material which has contraction coefficient $\gamma$, we should have $r(s)/r \geq \gamma$, and so we must have

$$\frac{\sin (s/R)}{s/R} \geq \gamma. \tag{8.4}$$

If $\gamma < 1$, then this inequality is satisfied at $s = 0$ and at sufficiently small values of $s$, but it puts an upper bound $F(\gamma)$ on $s/R$ defined by the equality $\sin (F(\gamma))/F(\gamma) = \gamma$, where $F$ is the inverse function of $t \mapsto \sin t/t$ (see the footnote[3]). This shows that the radius of $D$ can be at most $F(\gamma)R$. Therefore, the area of $\varphi(D)$ on the sphere will be at most

$$A(\gamma) = 2\pi(1 - \cos F(\gamma))R^2. \tag{8.5}$$

As the sphere $S_R^2$ of radius $R$ has Gaussian curvature $R^{-2}$, this shows that the integral of the curvature over $\varphi(D)$ is at most $A(\gamma)/R^2 = 2\pi(1 - \cos F(\gamma))$. This is a function of $\gamma$, independent of $R$. For $\gamma = 0$, it takes its maximum value $4\pi$.

For $\gamma = 0.5$ as in our experiment, we get $s/R \leq F(0.5) = 1.89$. Hence the tailoring method, which can at the most give the portion of $M$ whose area is $(1 - \cos F(\gamma))/2$ times the area of $S_R^2$, gives us $(1 - \cos (1.89))/2 = 0.65$ times the area of the sphere.

The interesting point is that any conceivable method of contraction moulding cannot achieve a better result in the sense of being able to mould a strictly larger portion of the above sphere. To see this, we argue by contradiction as follows. If possible, let $D'$ be another flat piece of plastic, with the same constant $\gamma$, which is contraction-moulded by some other hypothetical method to produce a part of $S_R^2$ that contains in its interior the part of $S_R^2$ produced by the above method. Let $\psi : D' \rightarrow S_R^2$ be the corresponding moulding function. By assumption $\psi(D')$ properly contains the portion of $S_R^2$ where $z \leq -R \cos (F(\gamma))$, which is the image of the disc of radius $F(\gamma)R$ by the function $\varphi$ used by the tailoring method. Hence there is a circle $C$ defined by $z = k_0$ on $S_R^2$ where $k_0 > -R \cos (F(\gamma))$, which is covered by the image of $\psi$. Note that the radius of $C$ is $\leq \sin F(\gamma)$, so the perimeter of $C$ is $\leq 2\pi \sin F(\gamma)$. Let $O' \in D'$ be the point that is mapped by $\psi$ to the south pole $(0, 0, -R) \in S_R^2$. As $\psi$ can only contract, the disc around $O'$ of radius $s$ in $D'$ has to go inside the portion of $S_R^2$ where $z = -R \cos (s/R) \leq s$, which is the image of the disc in $D$ of radius $s$ by the function $\varphi$ used by the tailoring method. Hence the inverse

---

[3]Consider the function $f : [0, \pi] \rightarrow [0, 1]$ defined by $f(0) = 1$ and $f(t) = \sin t/t$ for $t \neq 0$. This is a monotonically decreasing function, with $f(0) = 1$ and $f(\pi) = 0$. Hence it admits an inverse function $F : [0, 1] \rightarrow [0, \pi]$, with $F(0) = \pi$ and $F(1) = 0$.

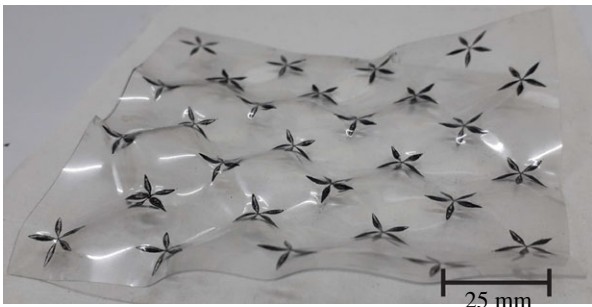

**Figure 12.** Patterned surface. The picture shows a textured plane, obtained by heating a flat piece of plastic which has a black and white pattern (can be doubly periodic as in this example). We can ensure that there is no large-scale bending by loosely sandwiching between two flat panes of glass while being exposed to radiation.

image $C' \subset D'$ of the circle $\psi(C)$ in $D'$ is a curve that lies entirely outside the circle in $D'$ of radius $F(\gamma)R$ centred at $O'$. This shows that the perimeter of the curve $C'$ is strictly greater that $2\pi F(\gamma)R$. Hence the contraction factor (length of $C$/length of $C'$) is strictly less that $\sin(F(\gamma))/F(\gamma) = \gamma$. This is physically impossible, as the contraction factor has to be $\geq \gamma > 0$.

## 8.2. Comparison between the moulding methods

A special feature of the tailoring method is that the painting pattern in tailoring usually involves long (system-sized) white bands. As these bands retain their lengths, this gives a degree of long-range control on the moulding process. This is quite unlike the other two methods which are based on the combined effects of a large number of local deformations arising out of a patterned lattice, where the statistical variations add up to produce greater uncertainties.

## 8.3. Moulding textured surfaces

Besides fashioning curved surfaces in $\mathbb{R}^3$, the method of selective heating and contraction can also be used to fashion textured planar surfaces. An example of this is shown in figure 12.

Data accessibility. The physical and thermal characterization of the material used in the paper are provided in electronic supplementary material figures S1–S3. The input patterns of all the shapes moulded in the paper are provided in the electronic supplementary material. The Matlab programs used to generate the various printed patterns are available for download at https://github.com/harshjn/GeometricMoldingByContraction.

Authors' contributions. H.J. performed the experiments. N.N. formulated the theory. S.G. and H.J. designed the experiments. N.N. and S.G. wrote the paper.

Competing interests. We declare we have no competing interests.

Funding. We acknowledge support of the Department of Atomic Energy, Government of India, under project no. 12-R&D-TFR-5.10-0100.

Acknowledgements. We thank Subhojoy Gupta for a useful discussion on differential geometry, Salman Alam for his help with the initial experiments.

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
