## [Reviewer comments · Royal Society Open Science]

Review History

RSOS-200011.R0 (Original submission)

Review form: Reviewer 1 (Kan Li)

Is the manuscript scientifically sound in its present form?

Yes

Are the interpretations and conclusions justified by the results?

Yes

Is the language acceptable?

Yes

Do you have any ethical concerns with this paper?

No

Have you any concerns about statistical analyses in this paper?

No

Recommendation?

Accept as is

Comments to the Author(s)

Most of the comments have been successfully addressed. The manuscript is now in a clear flow of story. I thank the authors for their hard work. I recommend this paper to be accepted as is.

Review form: Reviewer 2**Is the manuscript scientifically sound in its present form?**

Yes

Are the interpretations and conclusions justified by the results?

Yes

Is the language acceptable?

Yes

Do you have any ethical concerns with this paper?

No

Have you any concerns about statistical analyses in this paper?

No

Recommendation?

Accept with minor revision (please list in comments)

Comments to the Author(s)

The reviewer's technical comments are well addressed by the authors. The reviewer recommends its acceptance for publication.

One minor comment on the cited references. Some recently closely related work on shape shifting from 2D to 3D using the shrink paper are recommended to cite:

- 1) Q. Zhang et al., *Extreme Mechanics Letter*, 11, 111-120, 2017
- 2) Russell W Mailen et al 2019 *Smart Mater. Struct.* 28 045011

These two papers all discussed the realization of different curvatures through the patterning of ink patterns in shrink paper.

Decision letter (RSOS-200011.R0)

31-Jan-2020

Dear Dr Ghosh

On behalf of the Editors, I am pleased to inform you that your Manuscript RSOS-200011 entitled "Molding 3D curved structures by selective heating" has been accepted for publication in *Royal Society Open Science* subject to minor revision in accordance with the referee suggestions. Please find the referees' comments at the end of this email.

The reviewers and handling editors have recommended publication, but also suggest some minor

revisions to your manuscript. Therefore, I invite you to respond to the comments and revise your manuscript.

- Ethics statement

- Data accessibility

If you wish to submit your supporting data or code to Dryad (<http://datadryad.org/>), or modify your current submission to dryad, please use the following link:
<http://datadryad.org/submit?journalID=RSOS&manu=RSOS-200011>

- Competing interests

- Authors' contributions

- Acknowledgements

- Funding statement

Please ensure you have prepared your revision in accordance with the guidance at <https://royalsociety.org/journals/authors/author-guidelines/> -- please note that we cannot publish your manuscript without the end statements. We have included a screenshot example of

the end statements for reference. If you feel that a given heading is not relevant to your paper, please nevertheless include the heading and explicitly state that it is not relevant to your work.

Because the schedule for publication is very tight, it is a condition of publication that you submit the revised version of your manuscript before 09-Feb-2020. Please note that the revision deadline will expire at 00.00am on this date. If you do not think you will be able to meet this date please let me know immediately.

If your manuscript is newly submitted and subsequently accepted for publication, you will be asked to pay the article processing charge, unless you request a waiver and this is approved by Royal Society Publishing. You can find out more about the charges at <https://royalsocietypublishing.org/rsos/charges>. Should you have any queries, please contact openscience@royalsociety.org.

on behalf of Prof R. Kerry Rowe (Subject Editor)
openscience@royalsociety.org

Associate Editor Comments to Author:

The reviewers are broadly satisfied that your paper is ready for acceptance, though a couple of minor comments remain. While you should consider adding the recommended references, please only do so if there is a valid scientific reason for doing so.

Reviewer comments to Author:

Reviewer: 1

Comments to the Author(s)

Most of the comments have been successfully addressed. The manuscript is now in a clear flow of story. I thank the authors for their hard work. I recommend this paper to be accepted as is.

Reviewer: 2

Comments to the Author(s)

The reviewer's technical comments are well addressed by the authors. The reviewer recommends its acceptance for publication.

One minor comment on the cited references. Some recently closely related work on shape shifting from 2D to 3D using the shrink paper are recommended to cite:

1) Q. Zhang et al., *Extreme Mechanics Letter*, 11, 111-120, 2017

2) Russell W Mailen et al 2019 *Smart Mater. Struct.* 28 045011

These two papers all discussed the realization of different curvatures through the patterning of ink patterns in shrink paper.

Author's Response to Decision Letter for (RSOS-200011.R0)

See Appendix A.

Decision letter (RSOS-200011.R1)

04-Feb-2020

Dear Dr Ghosh,

It is a pleasure to accept your manuscript entitled "Molding 3D curved structures by selective heating" in its current form for publication in Royal Society Open Science.

Kind regards,
Lianne Parkhouse
Editorial Coordinator
Royal Society Open Science
openscience@royalsociety.org

on behalf of the Associate Editor and Professor R. Kerry Rowe (Subject Editor)
openscience@royalsociety.org

Appendix A

To
The Editor,
Open Science

Dear Professor,

Subject: Resubmission of the manuscript “Molding 3D curved structures by selective heating”

We are resubmitting our paper “Molding 3D curved structures by selective heating” to your journal. We have added the two references suggested by the second reviewer. These are the 17th and the 18th reference of the modified manuscript.

In addition the revised manuscript also has two additional statements to address the journal’s policy for ethics issues and funding sources.

Sincerely,

Authors
Tata Institute of Fundamental Research
Mumbai-400005
India

Reviewer I

Most of the comments have been successfully addressed. The manuscript is now in a clear flow of story. I thank the authors for their hard work. I recommend this paper to be accepted as is.

We thank the reviewer for recommending the paper to be published.

Reviewer II

The reviewer's technical comments are well addressed by the authors. The reviewer recommends its acceptance for publication.

One minor comment on the cited references. Some recently closely related work on shape shifting from 2D to 3D using the shrink paper are recommended to cite: 1) Q. Zhang et al., *Extreme Mechanics Letter*, 11, 111-120, 2017 2) Russell W Mailen et al 2019 *Smart Mater. Struct.* 28 045011 These two papers all discussed the realization of different curvatures through the patterning of ink patterns in shrink paper.

We thank the reviewer for recommending the paper to be published. In response to the above comment we have added the two references suggested by the reviewer. These are the 17th and the 18th reference of the modified manuscript. The change is

Page 2 of the manuscript

“Some earlier experiments reported in the literature aimed at obtaining curved surfaces in \mathbb{R}^3 from flat surfaces relied on modifying the flat Riemannian metric of the starting planar surfaces [4-5,16-18].”

17. Zhang Q, Wommer J, ORourke C, Teitelman J, Tang Y, Robison J, Lin G, Yin J. 2017 Origami and kirigami inspired self-folding for programming three-dimensional shape shifting of polymer sheets with light. *Extreme Mechanics Letters* 11, 111120.

18. Mailen RW, Wagner CH, Bang RS, Zikry M, Dickey MD, Genzer J. 2019 Thermo-mechanical transformation of shape memory polymers from initially flat discs to bowls and saddles. *Smart Materials and Structures* 28, 045011.